# Endometrial receptivity and implantation require uterine BMP signaling through an ACVR2A-SMAD1/SMAD5 axis

Diana Monsivais [1,2,12✉], Takashi Nagashima[1,11], Renata Prunskaite-Hyyryläinen[3], Kaori Nozawa[1,2], Keisuke Shimada [4], Suni Tang[1], Clark Hamor[1], Julio E. Agno[1,2], Fengju Chen[5], Ramya P. Masand[1], Steven L. Young[6], Chad J. Creighton [5,7], Francesco J. DeMayo[8], Masahito Ikawa [4], Se-Jin Lee[9,10] & Martin M. Matzuk[1,2,12✉]

During early pregnancy in the mouse, nidatory estrogen (E2) stimulates endometrial receptivity by activating a network of signaling pathways that is not yet fully characterized. Here, we report that bone morphogenetic proteins (BMPs) control endometrial receptivity via a conserved activin receptor type 2 A (ACVR2A) and SMAD1/5 signaling pathway. Mice were generated to contain single or double conditional deletion of SMAD1/5 and ACVR2A/ACVR2B receptors using progesterone receptor (PR)-cre. Female mice with SMAD1/5 deletion display endometrial defects that result in the development of cystic endometrial glands, a hyperproliferative endometrial epithelium during the window of implantation, and impaired apicobasal transformation that prevents embryo implantation and leads to infertility. Analysis of *Acvr2a*-PRcre and *Acvr2b*-PRcre pregnant mice determined that BMP signaling occurs via ACVR2A and that ACVR2B is dispensable during embryo implantation. Therefore, BMPs signal through a conserved endometrial ACVR2A/SMAD1/5 pathway that promotes endometrial receptivity during embryo implantation.

[1] Department of Pathology & Immunology, Baylor College of Medicine, Houston, TX, USA. [2] Center for Drug Discovery, Baylor College of Medicine, Houston, TX, USA. [3] Faculty of Biochemistry and Medicine, University of Oulu, Oulu, Finland. [4] Research Institute for Microbial Disease, Osaka University, Osaka, Japan. [5] Department of Medicine, Baylor College of Medicine, Houston, TX, USA. [6] Department of Obstetrics and Gynecology, University of North Carolina, Chapel Hill, NC, USA. [7] Dan L. Duncan Comprehensive Cancer Center, Baylor College of Medicine, Houston, TX, USA. [8] National Institute of Environmental Health Sciences, Research Triangle Park, NC, USA. [9] Jackson Laboratory for Genomic Medicine, Farmington, CT, USA. [10] University of Connecticut School of Medicine, Department of Genetics and Genome Sciences, Farmington, CT, USA. [11] Present address: Hanakoganei Ladies Clinic, Tokyo, Japan. [12] These authors jointly supervised this work: Diana Monsivais, Martin M. Matzuk. ✉email: dmonsiva@bcm.edu; mmatzuk@bcm.edu

The endometrium is the mucosal lining of the uterus that is the first point of contact between an implanting embryo and its mother[1]. In the endometrium, various cell types are critical in establishing a pregnancy, endometrial epithelial cells participate in maternal–embryonic communication during implantation, and stromal cells transform into a secretory cell type (i.e., decidualize), with the important role of nurturing the growth and development of the early embryo. As a requisite step for pregnancy, implantation is a highly coordinated process that requires carefully synchronized maternal/embryonic communication[1]. In humans, the period of maximal endometrial receptivity to embryos (the window of implantation) is achieved 7–10 days after ovulation[1], whereas in mice, it occurs ~4–4.5 days post coitus (dpc). The window of implantation is characterized by molecular and histological changes such as inhibition of epithelial cell proliferation, and epithelial cell remodeling; these changes are carefully coordinated in a time-dependent manner by the steroid hormones, estrogen (E2), and progesterone (P4), and other growth factors[2–5]. In mice, nidatory E2 is secreted prior to implantation and induces the expression of several factors, including the leukemia inhibitory factor (LIF), which primes the endometrium for implantation[6]. Perturbations in the amount or timing of the nidatory E2 surge impair implantation by affecting endometrial gene expression[7]. Therefore, the temporal regulation of gene expression by E2 is a critical step in early pregnancy success.

The BMPs are a class of highly conserved members of the transforming growth factor β (TGFβ) family with important functions during development, morphogenesis, and reproduction[8–11]. BMPs signal via a heterotetrameric cell surface receptor complex composed of BMP type 1 (activin-like receptors 2, −3, or −6; ALK2/3/6) and type 2 (BMP receptor type 2, activin receptor type 2 A, −2B; BMPR2/ACVR2A/2B) receptors that transmit signals via the SMAD1/5 transcription factors[12]. In the uterus, in vivo studies have shown that conditional deletion of BMP2 or ALK2 results in female infertility owing to defects in the post-implantation process of stromal cell decidualization[8,13]. Conditional ablation of the BMP type 2 receptor, BMPR2, results in infertility owing to placental defects during mid-to-late gestation[14]. The uterine-specific roles of ACVR2A and ACVR2B during early pregnancy have not yet been analyzed owing to developmental defects caused by global deletion[15,16]. During the peri-implantation period, BMP signals in the endometrium are mediated via ALK3, and conditional ablation of ALK3 results in infertility due to impaired endometrial receptivity and defective embryo attachment[17]. Conditional inactivation of other BMP ligands also perturbs endometrial function; BMP7 ablation results in reduced fertility due to defects during implantation that affect mid-gestation[18]. BMPs are also implicated in human fertility and silencing of BMP2 and ALK2 in human endometrial stromal cells impairs in vitro decidualization[9,13,19]. Whole-exome sequencing of patients with recurrent implantation failure has also identified the presence of a damaging mutation in BMP7[20]. However, knowledge gaps remain regarding the signaling pathways that are activated downstream of the BMPs during the window of implantation. The cell surface receptor complexes that transmit BMP signals are also unknown, and the specific BMPs that signal during the window of implantation remain to be identified. The goal of these studies is to identify the uterine-specific roles of SMAD1 and SMAD5 and to determine the cell surface receptor complex that mediates BMP signaling during pregnancy. Our studies identify a link between BMP/ACVR2A/SMAD1/5 signaling and E2/P4 action in the endometrium during the window of implantation.

## Results

### Conditional deletion of SMAD1 and SMAD5 results in female infertility due to implantation defects.

Immunohistochemistry (IHC) showed that phosphorylated SMAD1/SMAD5 (pSMAD1/5) is dynamically expressed in the endometrium of wild-type (WT) mice during early pregnancy (Fig. 1a–h). pSMAD1/5 is strongly expressed in the luminal epithelium and stroma of the endometrium at 1.5 days post coitum (dpc) and 2.5 dpc (Fig. 1a–d); at 3.5 dpc, pSMAD1/5 expression is not readily detected in the luminal epithelium but is present in the underlying stroma and glandular epithelium (Fig. 1e, f). At 4.5 dpc, pSMAD1/5 is detected in both the luminal epithelium and in the decidualizing stroma (Fig. 1g, h). pSMAD1/5 expression is excluded from the primary decidualizing zone adjacent to the embryo (Fig. 1g, black lines). pSMAD1/5 staining was also observed in human endometrial biopsies obtained from women during the proliferative or mid-secretory phase of the menstrual cycle (Fig. 1i–n). Staining was observed in the glandular epithelium with weak stromal expression during the proliferative phase (Fig. 1i, k), whereas pSMAD1/5 staining became more pronounced in the decidualizing stromal cells during the mid-secretory phase (yellow arrows, Fig. 1j–l).

Smad1 knockout (KO) mice are embryonically lethal at 9.5 dpc[21], whereas Smad5 KO mice also experience embryonic lethality due to defective embryonic and extraembryonic development[22]. Therefore, to determine the role of SMAD1 and SMAD5 during pregnancy, we utilized a conditional deletion approach using progesterone receptor-cre mice (Smad1$^{flox/flox}$-PRcre, "Smad1 cKO"; Smad5$^{flox/flox}$-PRcre, "Smad5 cKO"; or Smad1$^{flox/flox}$;Smad5$^{flox/flox}$-PRcre, "Smad1/5 cKO") to obtain SMAD1/5 deletion in PR-expressing tissues of the female reproductive tract[23]. Histological analysis of uteri from adult Smad1 cKO, Smad 5 cKO, and Smad1/5 cKO mice showed that all uterine layers were present and normally structured (Fig. 1o–v). A 6-month fertility trial indicated that conditional deletion of Smad1 resulted in normal fertility, conditional deletion of Smad5 resulted in subfertility, whereas double conditional deletion of Smad1/5 resulted in infertility (Fig. 1w, x and Supplementary Table 1). Timed mating analyses of Smad5$^{flox/flox}$-PRcre showed the presence of hemorrhagic implantation sites at 8.5 dpc (Supplementary Fig. 1a, b), with abnormal formation of the decidua in the dissected implantation sites (Supplementary Fig. 1c, d). Quantification of 8.5 dpc implantation sites of Smad5$^{flox/flox}$-PRcre females indicated that many implantation sites were resorbing (18/27 resorbing/normal, $n = 3$) compared with those of control mice (0/28 resorbing/normal, $n = 3$) (Supplementary Fig. 1e). Therefore, SMAD5 is critical for sustained decidualization and early pregnancy success. However, the decidualization defect in the Smad5$^{flox/flox}$-PRcre females was not completely penetrant, likely owing to SMAD1 compensation during decidualization.

Because Smad1/5 cKO mice were infertile and did not generate any pups over the course of the 6-month fertility trial, further studies were conducted on these mice to address the potential redundancy between SMAD1 and SMAD5 during pregnancy. We identified that effective deletion of both targeted exons in the Smad1 and Smad5 alleles was obtained in the uterine and ovarian tissues (Supplementary Fig. 1f, g). This corresponded to undetected pSMAD1/5 expression by IHC in 4.5 dpc implantation sites and by western blot (Supplementary Fig. 1h–j). Ovarian histology of randomly cycling 12-week-old control and Smad1/5 cKO mice showed normal structure, follicles, and corpora lutea (Supplementary Fig. 1k, l). Superovulation studies were performed to assess ovarian function independently of uterine function in the Smad1/5 cKO mice. Analysis of ovarian function showed that the Smad1/5 cKO females ovulated in response to pregnant mare serum gonadotropin (PMSG) + human chorionic gonadotropin (hCG) and that there was no difference in the serum levels of E2 or P4 (Supplementary Fig. 1m–o). Therefore, the ovarian function was normal in Smad1/5 cKO mice.

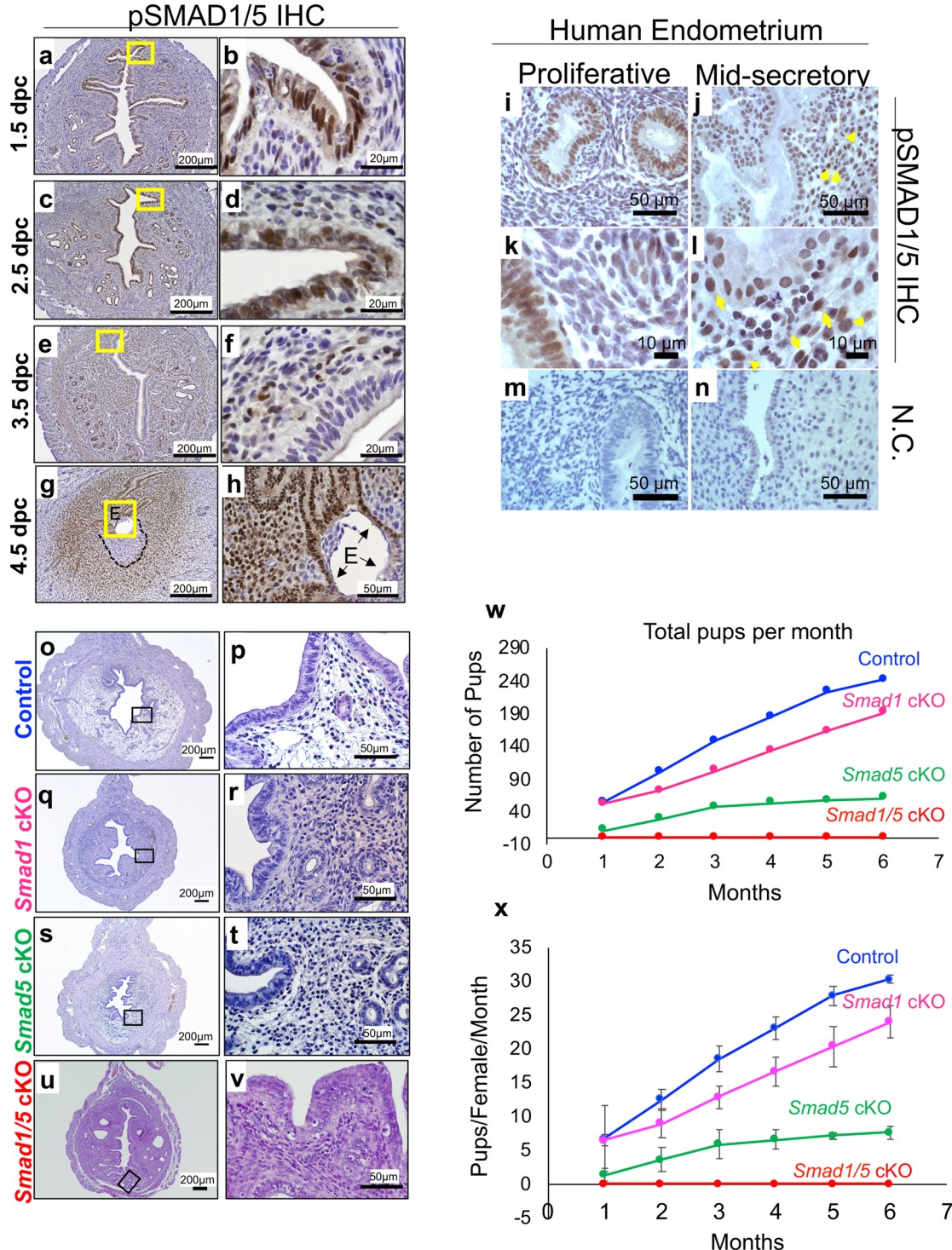

**Fig. 1 Conditional deletion of SMAD1/5 results in female infertility. a–h** pSMAD1/5 immunohistochemistry (IHC) at 1.5 dpc **a–b**, 2.5 dpc **c–d**, 3.5 dpc **e–f**, and 4.5 dpc (black dotted lines indicate primary decidual zone) **g–h**. *E*, embryo. Representative image of the embryo, observed in at least three specimens from different mice. Images in **a–h** represent findings observed in at least three individual samples per pregnancy timepoint. **i–n** pSMAD1/5 IHC in human endometrial biopsies obtained during the proliferative phase **i**, **k** or mid-secretory **j**, **l** of the menstrual cycle. **m**, **n** are negative controls, yellow arrows in **j**, **l** point to positive decidualized cells in the mid-secretory phase endometrium. Images shown are representative of patterns observed in three proliferative phase and six mid-secretory phase individuals. **o–v** H&E-stained uterine cross-sections from 12-week-old control **o–p**, *Smad1* cKO **q–r**, *Smad5* cKO **s–t,** and *Smad1/5* cKO **u–v** mice. **w–x** Fertility assessment in Control (*n* = 8), *Smad1* cKO (*n* = 8), *Smad5* cKO (*n* = 8), and *Smad1/5* cKO (*n* = 8) mice over the course of 6 months. Total pups per month are plotted in **w**, whereas the number of pups per female per month is plotted in **x**. Plotted as average pups per female per month ± standard error of the mean (SEM), *n* = 8 per genotype.

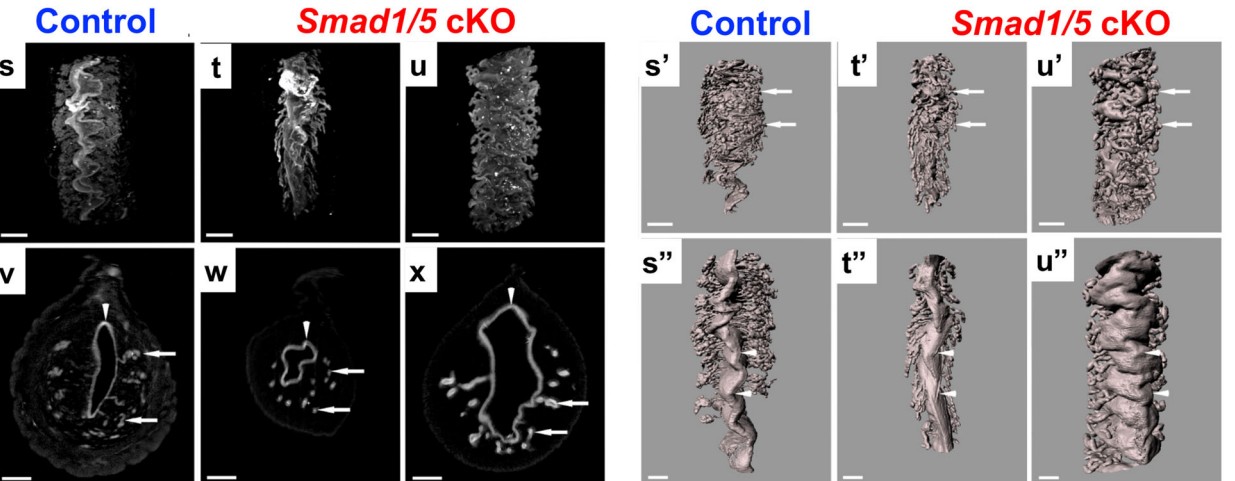

**SMAD1/5 signaling is essential for uterine gland 3D morphology and WNT-signaling.** Analysis of uterine morphology in control and *Smad1/5* cKO mice throughout development identified morphological defects in the uterine glands of the *Smad1/5* cKO mice that worsened with age (Fig. 2). IHC of the glandular-specific marker, FOXA2[24,25], in the 3- and 6-week-old uterus indicated the presence of glands in both control and *Smad1/5* cKO females (Fig. 2a–d). The glands enlarged and were observed

to be cystic at 6 weeks and 12-weeks of age, and became hemorrhagic at 24-weeks of age in the *Smad1/5* cKO mice (Fig. 2c–h). Quantitative PCR (qPCR) analysis demonstrated abnormal expression of the secreted frizzled receptor proteins (*Sfrp1-5*), which control WNT-signaling during endometrial glandular adenogenesis (Fig. 2i)[26].

Glandular defects of the *Smad1/5* cKO mice were assessed in three dimensions (3D) by whole-mount immunostaining of the

**Fig. 2 Smad1/5 cKO mice develop abnormal uterine glands that appear enlarged and cystic. a–h** Histological analysis of control (**a, c, e, g**) and *Smad1/5* cKO (**b, d, f, h**) uteri stained with FOXA2 (**a–d**) or H&E (**e–h**). Uteri were analyzed at 3 weeks (**a–b**), 6 weeks (**c–d**), 12 weeks (**e–f**), and 24 weeks of age (**g–h**). **i** Expression of the WNT-pathway inhibitors (*Sfrp1-5*) was analyzed using qPCR of 12-week-old uterine tissues of control (*n* = 3) and *Smad1/5* cKO (*n* = 3) mice. Histograms represent mean ± standard error of the mean (SEM), paired, two-tailed, *t* test, *$p < 0.05$, **$p < 0.01$, ***$p < 0.001$. *Sfrp1*, $p = 0.017$; *Sfrp2*, $p = 0.105$; *Sfrp3*, $p = 0.007$; *Sfrp4*, $p = 0.0107$; *Sfrp5*, $p = 0.042$. **j–r** Whole-mount immunostaining with FOXA2 antibody followed by multiphoton microscopy (**j–o**) in the uteri of non-pregnant control (**j**), or *Smad1/5* cKO mice (**k**). **l–r** show analyses performed on individual glands from control (**l, l′, l″**) or *Smad1/5* cKO mice (**m, m′, m″**) and the corresponding quantification of the width, length, and density of the glands (**p–r**). Total fields counted for gland density analysis: *n* = 8 in three control mice and *n* = 18 in three *Smad1/5* cKO mice. Histograms represent mean ± standard error of the mean (SEM). Unpaired, two-tailed *t* test, *$p < 0.033$, **$p < 0.002$, ***$P < 0.001$. Size bar: **j–k** is 500 μm; **l–m″** is 30 μm. Arrow in **o** indicates enlarged cystic endometrial gland from *Smad1/5* cKO. **s–x** Uterine lumen and endometrial glands stained with E-cadherin antibody and scanned by Optical Projection Tomography (OPT). Control (**s, s′, s″, v**) and *Smad1/5* cKO (**t–u′, w–x**) whole-mount (**s–u″**) and optical cross-sections (**v–x**) are displayed. Arrows in **v–x** point to the uterine glands and arrowheads indicate the uterine lumen. **s′–u″**) Surface rendering of the mouse uterus emphasizes uterine glands (arrows in **s′–u′**) and folds in uterine lumen (arrowheads in **s″–u″**). Scale bar **s–u′** (500 μm), and **v–x, s″–u″** (300 μm). OPT scans and multiphoton imaging were performed in the tissues of at least three control and three *Smad1/5* cKO mice.

glandular (FOXA2) or uterine epithelium (E-cadherin) in adult 6-month-old mice, followed by multiphoton microscopy or optical projection tomography imaging (OPT) (Fig. 2j–u″). 3D imaging revealed that in control uteri, endometrial glands were compactly sprouting from the lumen towards the myometrium making an ~90° angle with the lumen, whereas the *Smad1/5* cKO presented a disorganized and dilated glandular structure (Fig. 2j, k). Each of the indicated glands was individually selected and analyzed (Fig. 2l–r); the glands of the *Smad1/5* cKO mice had increased width (Fig. 2l, m, p), decreased length (Fig. 2l′, m′, q) and decreased overall density (Fig. 2r). Even though *Smad1/5* cKO glands appeared visibly longer, individual gland measurement revealed that the control glands were 56% longer due to increased coiling (Fig. 2l′, m′, q). Taken together, these data indicate that BMP signaling via SMAD1/5 has a significant role in uterine glandular morphology in adult virgin mice.

**Conditional deletion of SMAD1/5 causes abnormal response to E2 and P4.** The window of implantation in mice occurs between 3.5 and 4.5 dpc and is characterized by a transition from an E2-dominant proliferative state to a P4-responsive state[1,3,27]. At 3.5 dpc, the luminal uterine epithelium of control mice had no Ki67-positive cells yet showed prominent stromal Ki67-reactivity, whereas the epithelium of *Smad1/5* cKO mice continued to proliferate and had fewer Ki67-positive stromal cells (Fig. 3a, b). Likewise, mucin 1 (MUC1), which is typically downregulated in the receptive epithelium, had stronger apical expression in the *Smad1/5* cKO mice compared to the controls (Fig. 3c, d). PR expression was equally detected in the control and *Smad1/5* cKO uterus (Fig. 3e, f). qPCR analysis of the control and *Smad1/5* cKO 3.5 dpc endometrial epithelium revealed upregulation of E2-responsive genes, lipocalin 2 (*Lcn2*), leukemia inhibitory factor (*Lif*), mucin 1 (*Muc1*) and of proliferation-associated genes, cyclin D1 (*Ccnd1*), minichromosome maintenance complex component 2 (*Mcm2*) and minichromosome maintenance complex component 7 (*Mcm7*) (Fig. 3g). Expression levels of the genes encoding ER (*Esr1*) and PR (*Pgr*) were similar (Fig. 3g).

To test the endometrial response to exogenous hormones, control, and *Smad1/5* cKO mice were ovariectomized and administered a series of E2 and P4 treatments as outlined in Supplemental Fig. 2a. Uterine tissues were collected from the mice 15 h after the last injection of E2+P4 and the uterine epithelium was isolated and analyzed for the expression of genes indicative of E2 response (Supplemental Fig. 2b). Compared to the controls, the endometrial epithelium of the *Smad1/5* cKO mice showed unopposed E2 response, with a significantly elevated expression of the genes encoding chloride channel accessory 3 (*Clca3*), lipocalin 2 (*Lcn2*), and lactoferrin (*Ltf*) (Supplemental Fig. 2b). There was a decreased trend in the expression of the

gene encoding the progesterone receptor (*Pgr*), though the difference was not statistically significant. Expression of the gene encoding the ERα, *Esr1*, was similar between the control and *Smad1/5* cKO mice. Histologically, we observed that, unlike the controls, the luminal uterine epithelium of the *Smad1/5* cKO mice contained Ki67-positive cells and that the uterine lumen failed to close in response to E2+P4 administration (Supplemental Fig. 2c–h). Therefore, uterine SMAD1/5 is important for receptivity during the window of implantation and SMAD1/5 conditional deletion results in enhanced E2 action in the endometrium.

**Impaired embryo implantation and decidualization in Smad1/5 cKO mice at 4.5 dpc.** Analysis of implantation identified several implantation sites in the control mice at 4.5 dpc (black arrows, Fig. 3h) but none in the *Smad1/5* cKO mice (Fig. 3i). Instead, unattached embryos were recovered in the *Smad1/5* cKO mice by flushing of the uterus. Histological analysis of the implantation sites in the control mice demonstrated embryos fully encapsulated by the luminal uterine epithelium, whereas *Smad1/5* cKO mice had unattached embryos floating within the uterine lumen (Fig. 3j, k). IHC of the receptivity marker FOXO1[28] was abnormally retained in the cytoplasm of the *Smad1/5* cKO mice, whereas nuclear expression was detected in controls (Fig. 3l, m). PR was also defectively retained in the *Smad1/5* cKO luminal epithelium (Fig. 3n–o). qPCR analysis of 4.5 dpc uteri showed abnormal expression of implantation markers in the *Smad1/5* cKO mice (Fig. 3p). *Hand2* was downregulated in the mutants (Fig. 3p and Supplementary Fig. 2i–l) and other decidualization-related genes such as bone morphogenetic protein 2 (*Bmp2*), prostaglandin-endoperoxide synthase 2 (*Ptgs2/Cox2*), and Wnt family member 4 (*Wnt4*), were also downregulated in the mutant uteri (Fig. 3p). *Lif* and *Esr1* were increased in the *Smad1/5* cKO mice (Fig. 3p). Increased expression of P4-regulated genes that are critical for paracrine endometrial communication was detected in the *Smad1/5* cKO mice, *Nr2f2* (COUP-TFII), *Ihh*, *Ptch1*, and *Smo* (Supplementary Fig. 2m–o). The canonical Ihh/COUP-TFII pathway directs paracrine communication in the endometrium that halts E2-mediated proliferation of the luminal epithelium during implantation[27,29].

To test whether the implantation defects observed in the *Smad1/5* cKO mice could be rescued by administration of an estrogen receptor (ER) antagonist, control and female mice were injected with Vehicle or ICI 182,780 on the morning of 3.5 dpc (Supplementary Fig. 3a). Because implantation in mice occurs at 4.5 dpc, implantation was assessed at 5.5 dpc in this group of mice. Control mice treated with Vehicle or ICI 182,780 demonstrated the presence of implantation sites (Supplementary

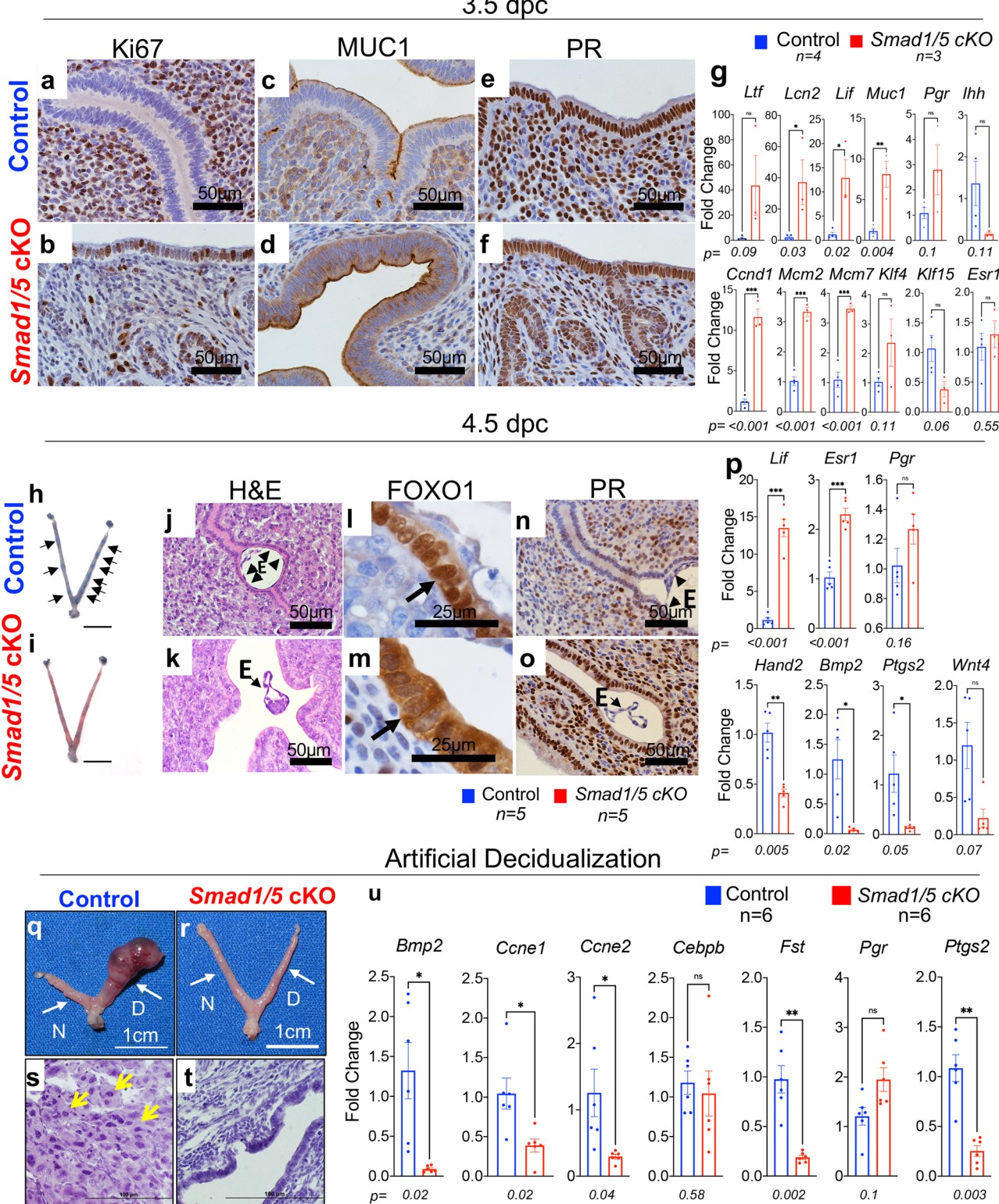

Fig. 3b, d). However, *Smad1/5* cKO mice did not show any implantation sites, even after ICI 182,780 administration (Supplementary Fig. 3c, e), indicating that antagonizing ER activity with ICI 182,780 during the window of implantation was not sufficient to rescue the implantation defects in the mutant mice. Quantification of the implantation sites in the control and mutant mice is shown in Supplementary Fig. 3f, and histological analyses of the control implantation sites and mutant uteri are presented in Supplementary Fig. 3g, n.

The uterine response to artificial decidualization[30] showed that unlike the control mice, whose uteri increased in size when injected with oil, the uteri of *Smad1/5* cKO mice failed to respond (Fig. 2q, r). Histology confirmed that only the stromal cells of the control mice decidualized (Fig. 2s, t). Genes involved in decidualization were decreased in the uteri of *Smad1/5* cKO mice (Fig. 2u). Together, these results indicate that SMAD1/5 signaling is critical for endometrial receptivity, implantation, and stromal cell decidualization.

**Fig. 3 Abnormal endometrial receptivity at 3.5 dpc and defective embryo implantation at 4.5 dpc in *Smad1/5* cKO mice. a–f** Histological analysis of 3.5 dpc uteri from control (**a**, **c**, **e**) and *Smad1/5* cKO (**b**, **d**, **f**) mice was performed by staining with Ki67 (**a–b**), MUC1 (**c–d**), or progesterone receptor (PR) (**e–f**). **g** qPCR analysis from the endometrial epithelium of control (blue bars) or *Smad1/5* cKO (red bars) mice. Paired, two-tailed, *t* test, mean ± SEM, *$p < 0.05$, **$p < 0.001$, ***$P < 0.0001$. **h–i** Whole uteri of control (**h**) and *Smad1/5* cKO (**i**) mice isolated at 4.5 dpc after injection with Chicago Sky Blue dye; implantation sites in the control mice can be visualized as blue bands (indicated by black arrows). Size bar = 1 cm. **j–k** H&E-stained cross-sections from 4.5 dpc control (**j**) and *Smad1/5* cKO (**k**) uteri. e=embryo. **l–o** IHC of FOXO1 **i–m** and progesterone receptor (PR) **n–o** in 4.5 dpc uterine cross-sections of control (**l–n**) and *Smad1/5* cKO (**m–o**) mice. Arrows in **l**, **m** indicate the nuclear staining of FOXO1 in the controls (**l**) and cytoplasmic FOXO1 staining in the *Smad1/5* cKO mice (**m**) *E* embryo. Representative image of the embryo, observed in at least three specimens from different mice. **p** qPCR analysis of implantation-related markers in control (blue bars, $n = 5$) and *Smad1/5* cKO (red bars, $n = 5$) uterine tissues collected at 4.5 dpc. Histology and qPCR analyses were performed in at least three samples of each genotype. Histograms represent mean ± SEM. Paired, two-tailed, *t* test, *$p < 0.033$, **$p < 0.002$, ***$P < 0.001$. **q–u** Analysis of decidualization reveals that compared with controls, *Smad1/5* cKO cannot respond to the artificial induction of decidualization. Gross images of control (**q**) and *Smad1/5* cKO (**r**) uteri, (*D*, decidual horn; *N*, non-decidual horn; indicated by white arrows). **s–t** H&E stains of uterine cross-sections of the decidual horns of control (**s**) and *Smad1/5* cKO (**t**) mice. Yellow arrows in (**s**) indicate decidualized cells. **u** Gene expression analysis by qPCR of control (blue bars, $n = 6$) and *Smad1/5* cKO (red bars, $n = 6$) uterine tissues that received the decidual stimulus. Histology and qPCR analyses were performed in at least three samples of each genotype. Histograms represent mean ± SEM. Paired, two-tailed, *t* test, *$p < 0.033$, **$p < 0.002$, ***$P < 0.001$.

**Conditional deletion of ACVR2A and ACVR2B results in female fertility defects.** BMPs signal via a heterotetrameric cell surface receptor complex that is composed of two BMP type 1 receptors (ALK2/ALK3/ALK6) and two BMP type 2 receptors (ACVR2A/ACVR2B/BMPR2) that activate intracellular signaling via the SMAD1/5 transcription factors[12]. BMP and activin-induced signaling is controlled by secreted protein antagonists, Noggin (BMP selective) and Follistatin (activin selective), which sequester the ligands and prevent the formation of an active signaling receptor complex[31–33]. It was previously determined that ALK3 is the type 1 receptor responsible for mediating BMP signals during implantation[17]; however, it is not known which BMP type 2 receptor is partnering with ALK3 in this process. Expression of the three BMP type 2 receptors, *Acvr2a, Acvr2b,* and *Bmpr2*, was measured in isolated endometrial stromal and epithelial tissues of WT mice that were ovariectomized and treated with E2 and P4 to simulate early pregnancy. qPCR analysis detected the presence of all three transcripts of epithelial and stromal cells of the endometrium (Supplementary Fig. 4a). *Bmpr2* was enriched in stroma compared with epithelium; whereas *Acvr2a* and *Acvr2b* were equally detected in both epithelium and stroma (Supplementary Fig. 4a). RNAseq data from WT mouse uterus at 3.5 dpc of pseudopregnancy also showed that all three receptors were expressed, with *Bmpr2* and *Acvr2a* being more abundant than *Acvr2b* (Supplementary Fig. 4b). Given that a previous mouse model with conditional inactivation of BMPR2 supported implantation[14], we hypothesized that BMPs were likely signaling via ACVR2A and/or ACVR2B during early pregnancy.

To overcome the perinatal lethality observed in ACVR2A[15] and ACVR2B[16] null mice, and to study their roles in the reproductive tract, we generated mice with single conditional deletion of ACVR2A and ACVR2B using PRcre (*Acvr2a*$^{flox/flox}$-PRcre "*Acvr2a* cKO" and *Acvr2b*$^{flox/flox}$-PRcre "*Acvr2b* cKO") (Supplementary Fig. 5a, i). Deletion of the targeted exons was verified by qPCR in each of the genotypes (Supplementary Fig. 5a–c, i–j). A 6-month fertility trial indicated that *Acvr2a* cKO females were infertile and did not generate any pups, whereas *Acvr2b* cKO females were subfertile (56.4 ± 12.74 pups/female vs. 26.6 ± 9.33 pups/female in control vs. *Acvr2b* cKO, $n = 10$ each) (Fig. 4a–b, Supplementary Table 1). Superovulation experiments showed no significant differences in the total number of ovulated oocytes in either *Acvr2a* cKO (30.2 ± 10.26 vs. 12.42 ± 5.96, $p = 0.13$) or *Acvr2b* cKO mice (51.83 ± 4.98 vs. 56.4 ± 7.73, $p = 0.62$) (Supplementary Fig. 5h, m). Though not statistically significant, there was a reduction in the number of ovulated oocytes in the *Acvr2a* cKO mice, and some mice did not respond to super-ovulation (Supplementary Fig. 5h). To study the morphology of

the ovary, mice were induced to superovulate with PMSG plus hCG for 6 h (to assess pre-ovulatory follicles). Morphological analysis of the ovaries revealed both normally developing follicles and follicles with defective cumulus cells (Supplementary Fig. 5d, e). Corpora lutea were analyzed 18 h after PMSG plus hCG administration, and analyses showed the presence of corpora lutea in the ovaries of both control and *Acvr2a* cKO mice (Supplementary Fig. 5f, g). PR is briefly expressed in the granulosa cells of post-ovulatory follicles[23]. Ovarian histology of *Acvr2b* cKO mice, on the other hand, showed no defects and had follicles at various stages (Supplementary Fig. 5k, l). Thus, although *Acvr2b* cKO mice showed normal ovarian architecture and function, analysis of the *Acvr2a* cKO ovaries showed that despite having the ability to ovulate and form a corpus luteum, the ovarian function was not completely normal.

Analysis of control and *Acvr2b* cKO mice at 4.5 dpc revealed no difference in the number (control $n = 8$, 8.12 ± 0.61 vs. *Acvr2b* cKO $n = 7$, 7.8 ± 0.66) or weight (control $n = 4$, 11.7 mg ± 1.33 vs. *Acvr2b* cKO $n = 3$, 9.68 mg ± 0.5) of the implantation sites. At 10.5 dpc, resorbing implantation sites were identified in *Acvr2b* cKO mice that were reduced in weight compared to implantation sites of control mice (control $n = 4$, 13.5 mg ± 0.62 vs. *Acvr2b* cKO $n = 4$, 9.85 mg ± 0.99, $p = 0.018$) (Fig. 4c, d). Histological analysis of the implantation sites revealed the presence of a hemorrhagic decidual layer, absence of uterine natural killer cells (black arrowheads), and the abnormal expansion of trophoblast giant cells (black arrows) in *Acvr2b* cKO mice (Fig. 4e, f). Uterine natural killer cells are recruited to the decidua of developing embryos and are critical for remodeling of the placental vasculature[34]. Induction of artificial decidualization (Fig. 4g) determined that there was no difference between control and *Acvr2b* cKO mice, and decidualization progressed equally in both genotypes (Fig. 4h, i). Thus, the uterus of *Acvr2b* cKO mice supports implantation, and subfertility appears secondary to post-implantation abnormalities.

**Analysis of implantation, estrous cycles, fertilization, and pre-implantation embryo development in *Acvr2a* cKO mice.** IHC analysis of ACVR2A in non-pregnant and 4.5 dpc implantation sites (Fig. 5a, f) determined that ACVR2A is expressed in the decidualizing endometrial stroma and epithelium during the implantation process, suggesting a critical role for ACVR2A during early pregnancy. Timed mating analysis of control and *Acvr2a* cKO females at 4.5 dpc revealed that unlike the control females, which showed numerous implantation sites (Fig. 5g, black arrows), *Acvr2a* cKO females had no implantation sites (Fig. 5h). Because activins may also signal via the ACVR2A/SMAD2/3 signaling pathway, which controls

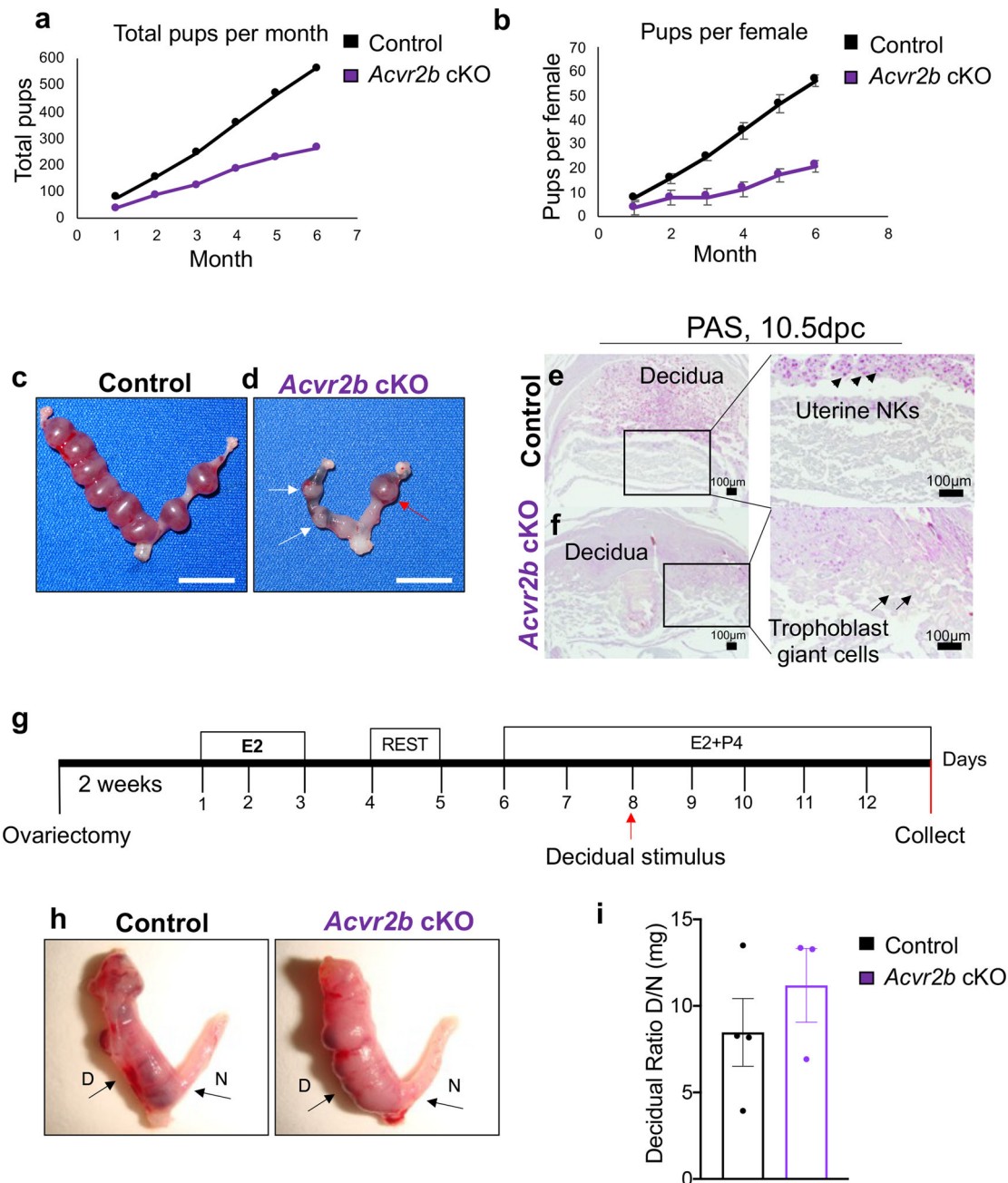

**Fig. 4 *Acvr2b* cKO mice are subfertile due to mid-gestation defects. a–b** Fertility assessment in the control (*n* = 10) and *Acvr2b* cKO (n = 10) mice over the course of 6 months. Total pups per month are plotted in **a**, while the number of pups per female per month is plotted in **b**. Data in **b** represent mean ± standard deviation, analyzed by single-factor ANOVA, *p* = 0.033. **c–d** Gross uterine images of 10.5 dpc implantation sites of control (**c**) and *Acvr2b* cKO (**d**) mice. Arrows in **d** indicate hemorrhagic and resorbing implantation sites. Red arrow indicates the implantation site analyzed by PAS in **f**. Size bar = 1 cm. **e–f** PAS-stained cross-sections from 10.5 implantation sites of control (**e**) and *Acvr2b* cKO (**f**) mice. Uterine natural killer cells (uNKs) are visualized as pink structures (indicated by black arrowheads) in the control implantation sites (**e**) but are absent in the *Acvr2b* cKOs (**f**). Abnormally expanded trophoblast giant cells are observed in the implantation sites of *Acvr2b* cKO mice (indicated by black arrows). **g** Experimental scheme used to induce artificial decidualization in control and *Acvr2b* cKO mice. **g** Gross images and quantification of the uterine horn weights after a 5-day decidualization. *D*, decidual horn; *N*, non-decidualized horn. **i** Decidual ratio was quantified by calculating the weights of the decidual horn relative to the control non-decidualized horn for each mouse. Images are representative of experiments performed in at least three subjects of each genotype. Histogram in **i** represents mean ± SEM. Unpaired, two-tailed *t* test, *p* = 0.40.

the hypothalamic-pituitary gonadal axis[35], we assessed whether *Acvr2a* cKO females were undergoing estrous cycles. Cytological examination of daily vaginal smears in control (*n* = 4) and *Acvr2a* cKO (*n* = 5) mice showed that both groups were cycling and experienced an equal number of estrous cycles (Supplementary Fig. 6a). Analysis of E2 and P4 serum levels during the estrous phase,

and follicle-stimulating hormone (FSH) during diestrus, showed no significant differences between the genotypes (Supplementary Fig. 6b–d). Therefore, the *Acvr2a* cKO females were undergoing estrous cycles and producing E2, P4, and FSH at normal levels.

Analysis of fertilization and blastocyst development was performed by tracking fertilized eggs from superovulated control

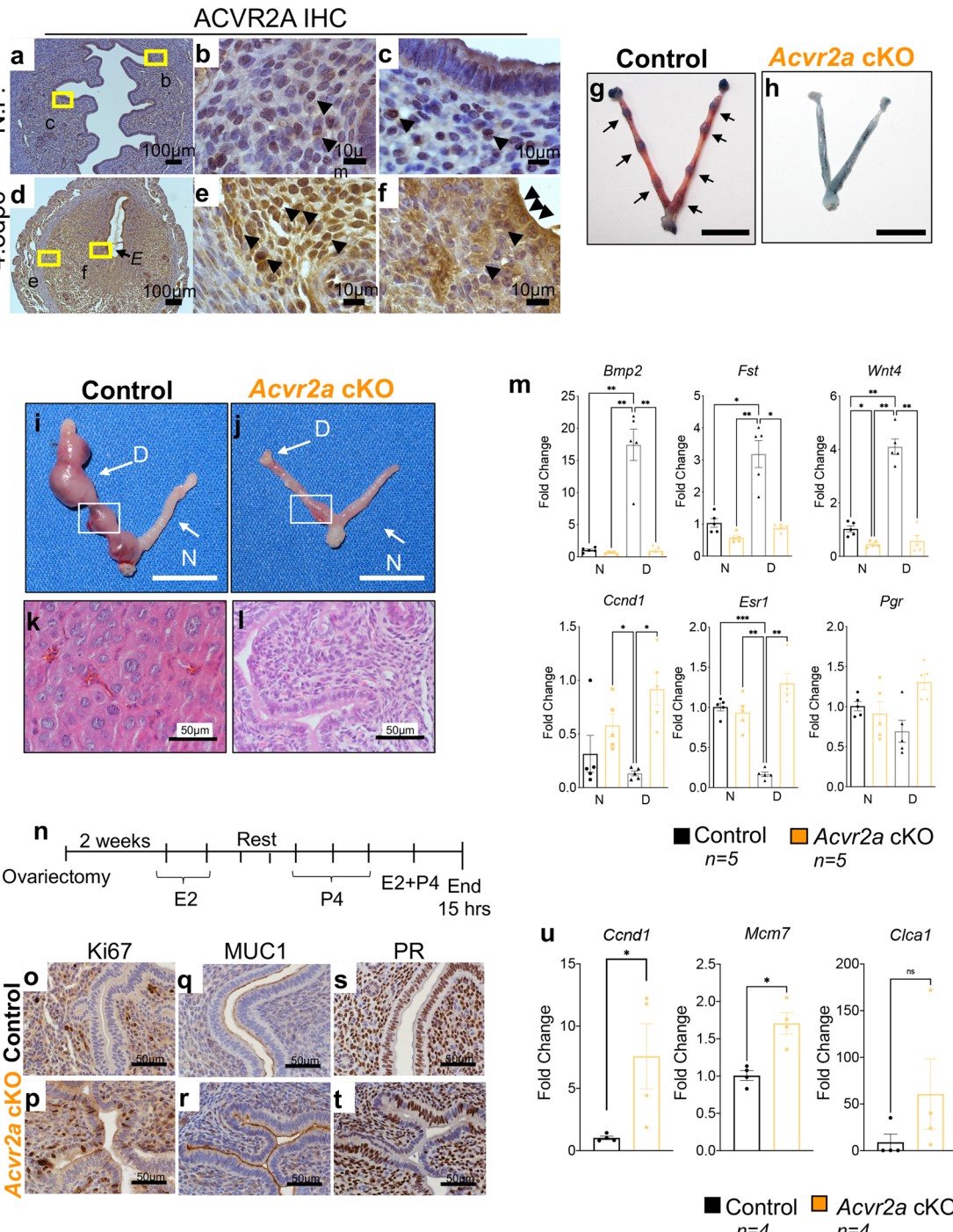

**Fig. 5 *Acvr2a* cKO mice are infertile and show defective endometrial receptivity. a–f** ACVR2A immunohistochemistry (IHC) was performed in mouse uterine cross-sections in non-pregnant (**a–c**) and 4.5 dpc pregnant **d–f** WT mice. *E*, denotes embryo; black arrowheads indicate positively-stained cells. **g–h** Images of 4.5 dpc uteri of control (**g**) and *Acvr2a* cKO (**h**) mice injected with Chicago Sky Blue dye to visualize implantation sites (denoted by black arrows in **g**). Size bar = 1 cm. Decidual response was measured in control (**i**) and *Acvr2a* cKO (**j**) mice. *D*, decidual horn; *N*, non-decidual horn; Size bars = 1 cm. **k–l** H&E-stained cross-sections of the decidualized horns of control (**k**) and *Acvr2a* cKO (**l**) mice. **m** qPCR analysis of decidual-related genes, *Bmp2* (*p = 0.002), Fst (p = 0.002), Wnt4 (p < 0.0001), Ccnd1 (p = 0.03), Esr1 (p = 0.0007)* and *Pgr (p = 0.529)* in the non-decidual (*N*) and decidualized (*D*) tissues of control (black bars, *n* = 5) and *Acvr2a* cKO (orange bars, *n* = 5) mice. Histograms represent mean ± SEM. Analyzed by one-way ANOVA with Tukey's multiple comparisons post-test, asterisks above each bar represent statistically significant difference, *$p < 0.033$, **$p < 0.002$, ***$P < 0.001$. **n** Experimental scheme used to test uterine response to steroid hormones (Pollard experiment). **o–t** IHC of uterine tissues from control (**o, q, s**) and *Acvr2a* cKO mice (**p, r, t**), stained with Ki67 (**o, p**), MUC1 (**q, r**), and PR (**s, t**). **u** qPCR quantification of *Ccnd1 (p = 0.046), Mcm7 (p = 0.004)* and *Clca3 (p = 0.23)* in the uteri of control (black bars, *n* = 4) and *Acvr2a* cKO (orange bars, *n* = 4) mice collected 15 h after 10 ng E2 + 1 mg P4 administration. Images represent experiments performed in at least three subjects of each genotype. Histograms **u** represent mean ±SEM. Paired, two-tailed *t* test, *$p < 0.033$, **$p < 0.002$, ***$P < 0.001$.

and *Acvr2a* cKO mice mated to WT males (Supplementary Fig. 6e, f). The number of fertilized eggs was quantified the morning after mating (0.5 dpc), and daily monitoring of the development of zygotes to the two-cell and blastocyst stages occurred at comparable rates in both genotypes (Supplementary Fig. 6g). Therefore, oocyte fertilization and early embryo development to the blastocyst stage were normal in *Acvr2a* cKO mice.

**Endometrial receptivity and decidualization defects in *Acvr2a* cKO mice**. To assess the uterine function of *Acvr2a* cKO mice, we performed an artificial induction of decidualization (described in Fig. 4g). The stimulated uterine horn of the *Acvr2a* cKO mice did not decidualize (Fig. 5i, j). Histology of the stimulated uterine horn from control mice showed enlarged, cuboidal decidual cells, whereas those from the *Acvr2a* cKOs were unchanged (Fig. 5k, l). Unlike the controls, decidual-related gene expression (*Bmp2, Fst,* and *Wnt4*) did not increase in the *Acvr2a* cKO mice (Fig. 5m). Compared to the decidual horns of WT mice, *Ccnd1* and *Esr1* were upregulated in the decidual horns of *Acvr2a* cKOs (Fig. 5m), however, no difference was detected in *Pgr* expression. These results indicated impaired decidualization in the uterus of *Acvr2a* cKO mice.

To test the endometrial response to the steroid hormones, ovariectomized control and *Acvr2a* cKO mice were treated with E2 and P4 to mimic early pregnancy (Fig. 5n). Unlike the luminal uterine epithelium of the controls, the epithelium of *Acvr2a* cKO mice continued proliferating (presence of Ki67-positive cells) in response to P4 (Fig. 5o, p). Increased MUC1 expression was detected in the apical region of the luminal epithelium of *Acvr2a* cKO mice (Fig. 5q, r) and PR was decreased in the epithelium of the *Acvr2a* cKO mice (Fig. 5s, t). qPCR analysis of the uterus showed that *Acvr2a* cKO mice expressed increased levels of the proliferative markers, *Ccnd1* and *Mcm7*, while the E2-responsive gene, *Clca3*, was elevated but not significantly (Fig. 5u). Overall, ACVR2A has a crucial role during early pregnancy in the uterus, conferring normal hormonal response and preparing the endometrium for blastocyst attachment and decidualization.

**Assessing implantation defects in *Smad1/5* cKO and *Acvr2a* cKO mice by performing embryo transfers**. To confirm that defective endometrial receptivity contributed to the implantation failure observed in the *Smad1/5* cKO and *Acvr2a* cKO mice, WT embryos derived from WT donors were transferred to the uterine lumen of pseudopregnant control (n = 6), *Smad1/5* cKO (n = 3), or *Acvr2a* cKO (n = 4) recipient females (Supplementary Fig. 7a–e). Forty-eight hours after embryo transfers (equivalent to 5.5 dpc), several implantation sites were identified in the control recipient female mice (7.4 ± 2.3 implantation sites per mouse, n = 6), however, no transferred embryos implanted into *Smad1/5* cKO or *Acvr2a* cKO recipients (Supplementary Fig. 7b–e). To rule out any potential embryonic contribution to the infertility phenotype of the *Smad1/5* cKO and *Acvr2a* cKO mutant mice, we also transferred embryos derived from *Smad1/5* cKO and *Acvr2a* cKO donors to the uteri of WT recipients. When transferred to the uteri of pseudopregnant WT females, embryos derived from control, *Smad1/5* cKO and *Acvr2a* cKO females, implanted at similar rates (Supplementary Fig. 7f–i). These results demonstrated that pregnancy failed owing to implantation defects in *Smad1/5* cKO and *Acvr2a* cKO mice.

**Conserved signaling pathways in SMAD1/5 and ACVR2A cKO mice prevent apicobasal remodeling during the window of implantation**. To verify that the control and mutant mice had the expected E2 and P4 expression levels during early pregnancy, we quantified serum hormone levels in non-pregnant and pseudopregnant mice at 3.5 dpc. As expected, we observed that the serum levels of P4 increased at 3.5 dpc relative to the non-pregnant state (Supplementary Fig. 8a). Serum E2 rises during the window of implantation, also known as the nidatory E2 surge; we observed a slight induction in serum E2 levels relative to the non-pregnant state (Supplementary Fig. 8b), however, the increase was not statistically significant, likely because the E2 surge is transient[36,37]. Despite showing no significant differences in the serum levels of E2 during the window of implantation relative to the controls, the endometrial epithelium of the mutant mice showed unopposed epithelial proliferation and unlike the controls, had numerous Ki67-positive stained cells (Supplementary Fig. 8c–e). To assess *Smad1, Smad5,* and *Acvr2a* gene expression patterns in the uterine tissues of the control and mutant mice, we quantified the levels of these transcripts in the uteri of 3.5 dpc pseudopregnant mice. Although *Acvr2a* gene expression levels rose significantly in the uterine tissues of *Smad1/5* cKO mice, neither *Smad1* nor *Smad5* increased significantly in the uteri of the *Acvr2a* cKO mice (Supplementary Fig. 8f–h). We also quantified the gene expression of the secreted SMAD signaling inhibitor, LEFTY1, and found no differences in its gene expression levels in the *Smad1/5* cKO or *Acvr2a* cKO mice (Supplementary Fig. 8i)[32,38].

RNAseq was used to determine the global transcriptional profiles of uterine tissues from 3.5 dpc pseudopregnant control (n = 3), *Smad1/5* cKO (n = 4), and *Acvr2a* cKO (n = 4) mice (Fig. 6). Hierarchical clustering of the gene expression profiles identified that a large number of transcripts were shared between the mutants and differentially expressed in the controls (Supplementary Fig. 9). Compared to the uteri of control mice, *Smad1/5* cKOs had 818 upregulated genes and 488 downregulated genes (>1.4-fold, <0.6-fold, p < 0.01), whereas *Acvr2a* cKO mice displayed 435 upregulated and 300 downregulated genes, (>1.4-fold, <0.6-fold, p < 0.01) (Fig. 6a, b, Supplementary Data 1). Of these, 197 upregulated and 118 downregulated genes were shared between *Smad1/5* cKO and *Acvr2a* cKO mice (Supplementary Data 2 and 3). Genes that were previously known to be involved in endometrial receptivity pathways were abnormally expressed in both genotypes, such as, heart- and neural crest derivatives-expressed protein 2 (*Hand2*)[5]; patched-1 (*Ptch1*) and nuclear receptor subfamily 2 group F member 2 (*Nr2f2/Coup*-TFII)[3,4,39]; epidermal growth factor receptor (*Egfr*)[40]; kruppel-like factor 15 (*Klf15*)[17,41]; the gene encoding the interleukin 15 receptor (*Il15ra*), which is crucial for uterine natural killer cell differentiation[42,43]; interleukin-13 subunit alpha-2 (*Il13ra2*)[44] (Supplementary Data 3 and Supplementary Fig. 8j–p). Genes associated with E2 response were elevated in the mutants, such as mucin 1 (*Muc1*), chloride channel accessory 1 (*Clca1*), lipocalin 2 (*Lcn2*) (Supplementary Fig. 8q–u).

We observed abnormal expression of gene families involved in the BMP and WNT/β-catenin signaling pathway, such as follistatin (*Fst*), noggin (*Nog*), gremlin 2 (*Grem2*), inhibin beta b (*Inhbb*), bone morphogenetic protein 4 (*Bmp4*), as well as several of the genes encoding the frizzled and secreted frizzled-related protein gene family (Supplementary Data 3). Several upregulated genes in the mutant mice uteri were involved in ciliated cell function such as nucleoside diphosphate kinase 3 (*Nme3*) and forkhead box J1 (*Foxj1*) and in apicobasal polarity/plasma membrane transformation[45,46] such as desmoplakin (*Dsp*), desmoglein (*Dsg2*), epithelial cellular adhesion molecule (*Epcam*), e-cadherin (*Cdh1*), ezrin (*Ezr*) and gap junction beta-2 (*Gjb2*)[47–49] (Fig. a–b, Supplementary Data 3). Distal-less homeobox-5 and −6 (*Dlx5/Dlx6*), which are important for glandular cell development were also upregulated in the mutant mice uteri (Supplementary Data 3)[50]. Several genes involved in iron

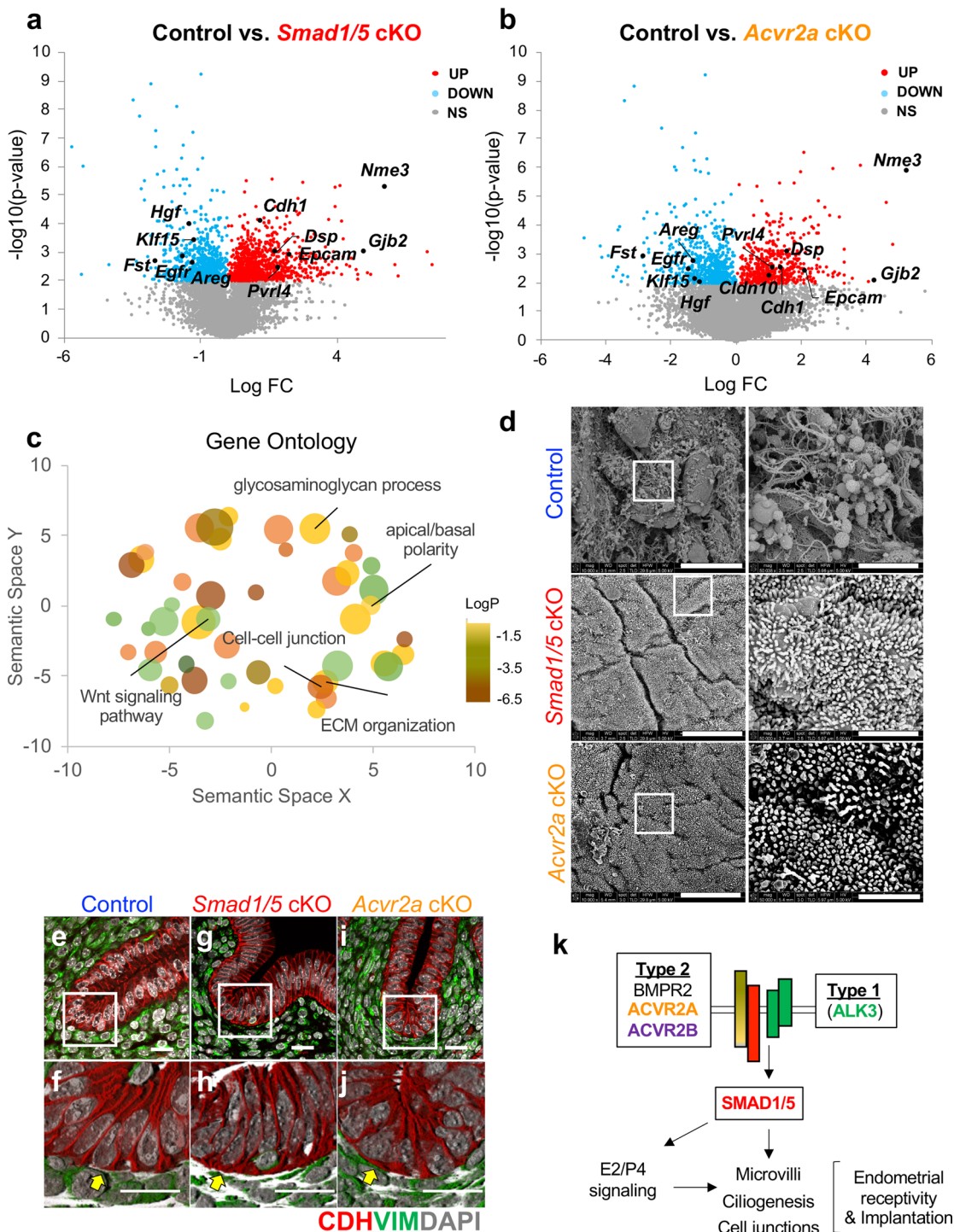

**Fig. 6 Shared signaling pathways between *Smad1/5* cKO and *Acvr2a cKO* mice reveal abnormal retention of apicobasal polarity and defective endometrial receptivity. a–b** Volcano plots of differentially expressed transcripts determined by RNAseq between control *vs. Smad1/5* cKO (**a**) and control vs. *Acvr2a* cKO (**b**) in the uterine tissues of 3.5 dpc pseudopregnant mice. Red, upregulated (fold change >1.4, *p* < 0.01 by paired, two-tailed, *t* test); blue, downregulated (<0.6, *p* < 0.01); labeled genes share differential expression in both genotypes. **c** Gene ontology classification of the shared genes in *Smad1/5* cKO and *Acvr2a* cKO mice that are differentially expressed vs. controls. Bubble size and colors are plotted relative to *p* values, whereas the location in the scatterplot represents functional categorization. **d** Scanning electron microscopy analysis of the surface of the luminal uterine epithelium of 3.5 dpc control, *Smad1/5* cKO, and *Acvr2a* cKO mice. Size bars = 10 μm (left) or 2 μm (right). Images in **d** are representative images obtained from analyses of three samples per genotype. **e–j** Immunofluorescence of the uterus of control (**e–f**), *Smad1/5* cKO (**g–h**) and *Acvr2a* cKO (**i–j**) mice at 3.5 dpc of pseudopregnancy. Tissues were stained with E-cadherin (red), vimentin (green), and DAPI (white), **e, g, i** are confocal z-stacks; **f, h, j**, are 3D-renderings of the z-stacks. Yellow arrows in the control uterus (**f**) indicate that E-cadherin immunoreactivity was decreased in the basal region of the luminal epithelium of control mice but maintained in the *Smad1/5* cKO (**h**) and *Acvr2a* cKO (**j**) mice. Size bars are 20 μm. **e–j** are representative images of at least three samples analyzed per genotype. **k** Schematic of the BMP signaling pathway that is active during the window of implantation.

transport[51,52] (Lipocalin 2 (*Lcn2*); Solute Carrier Family 40 Member 1, (*Slc40a1*); Scavenger Receptor Class A Member 5 (*Scara5*); Solute Carrier Family 7 Member 11 (*Slc7a11*)), retinoid and hormone transport and metabolism[53,54] (Cellular Retinoic Acid Binding Protein 2 (*Crabp2*); Cytochrome P450 Family 26 Subfamily A Member 1 (*Cyp26a1*); Cytochrome P450 Family 1 Subfamily B Member 1 (*Cyp1b1*)), suggesting that BMP/SMAD1/5 signaling is a critical regulator of these pathways in preparation for embryo implantation and receptivity. Revised gene ontology (GO) analysis[55] indicated that cellular processes involved in canonical Wnt-signaling, cell–cell junctions, apicobasal polarity, and extracellular matrix organization were highly overrepresented in the dataset of differentially expressed genes (Fig. 6c, Supplementary Data 3).

The surface of the luminal epithelium from control, *Smad1/5* cKO, and *Acvr2a* cKO mice were analyzed at 3.5 dpc of natural pregnancy using scanning electron microscopy (Fig. 6d). In the control mice, the presence of a remodeled receptive luminal epithelial surface was observed, with few microvilli and several secretory vesicles or "pinopodes" (Fig. 6d). Unlike the controls, both *Smad1/5* cKO and *Acvr2a* cKO mice lacked pinopodes, and had dense microvilli on the apical surface of the luminal epithelium (Fig. 6d). Confocal microscopy was used to visualize immunofluorescent staining of E-cadherin (epithelial marker, red) and vimentin (stromal marker, green) in the uterus of control, *Smad1/5* cKO and *Acvr2a* cKO mice at day 3.5 of pseudopregnancy (Fig. 6e–j). E-cadherin immunoreactivity was decreased in the basal region of the luminal epithelium (Fig. 6e, f, yellow arrow), corresponding to the loss of apical cell polarity that occurs during implantation. However, E-cadherin remained in the basal region of the luminal epithelium of *Smad1/5* cKO and *Acvr2a* cKO mice (Fig. 6g, j, yellow arrows). This confirmed RNAseq data, where the gene encoding E-cadherin (*Cdh1*), was more highly expressed in the *Smad1/5* cKO and *Acvr2a* cKO mice (Supplementary Data 3). Overall, transcriptomic and microscopy analyses indicated the presence of defective luminal epithelial remodeling in the mutant mice, as indicated by the maintenance of apicobasal cell polarity during the window of implantation (Fig. 6k). These results indicate that BMP signaling, occurring via an ALK3/ACVR2A/SMAD1/5 pathway, is critical for apicobasal remodeling of the endometrial epithelium that controls endometrial receptivity during the window of implantation.

## Discussion

We provide in vivo evidence underscoring the critical roles of a conserved ACVR2A/ALK3/SMAD1/5 signaling axis during the window of implantation. Previous studies from our group revealed that ALK3 is a critical BMP type 1 receptor required for endometrial receptivity in mice and that conditional deletion of ALK3 results in infertility owing to implantation defects[17]. The results presented herein phenocopy the uterine defects observed in the ALK3 mutant mice and provide essential in vivo evidence for the interaction of BMPs with ALK3 and ACVR2A during the window of implantation. Structural evidence exists for various BMP ligand/receptor interactions; for example, crystallography structures for BMP2/ACVR2A/ALK3 and BMP7/ACVR2A indicate BMP ligand/receptor specificity[56–59]. Our genetic findings provide in vivo evidence that the BMPs associate with ACVR2A and ALK3 to form an active cell surface receptor complex that is required for endometrial receptivity and embryo implantation. Using mice with conditional deletion of ACVR2B, we identified that signaling via this receptor was dispensable during implantation, and that even though the mice experienced subfertility, the defects arose at mid-gestation. Therefore, no further mechanistic studies were performed with the *Acvr2b* cKO mice during

implantation. Because previous studies showed that mice with conditional BMPR2 deletion could support implantation[14], we concluded that ACVR2A was the sole BMP type 2 receptor required for BMP signaling during the window of implantation. Analyses of ovarian function, independent of uterine function, were performed in these mouse models by administering PMSG plus hCG. Ovulation rate and ovarian architecture were normal in *Smad1/5* cKO and *Acvr2b* cKO mice, and subtle defects were observed in the ovaries of *Acvr2a* cKO mice.

The BMP ligands that signal via the ACVR2A/ALK3/SMAD1/5 pathway during the window of implantation are not yet known. Recent studies indicate that balanced BMP signaling is required during the window of implantation, as conditional deletion of follistatin, a potent activin inhibitor, perturbs implantation by leading to excessive activin expression, SMAD2/3 activation, and impaired embryo implantation[60]. *Bmp2* cKO mice are infertile, but their pregnancies fail post-implantation as a result of decidualization defects[8]. Mice with conditional BMP7 inactivation, on the other hand, are subfertile owing to peri-implantation defects that perturb mid-gestation development[18]. BMP5/7 DKO mice did not show any defects in fertility beyond those experienced by BMP7 deletion alone[18]. Recent evidence suggests that BMPs may function potently as heterodimers[61], therefore, future studies will be necessary to determine whether heterodimer signaling controls endometrial receptivity.

Additional mouse models have demonstrated the role of the BMP signaling pathway in the post-implantation period and point to the delicate balance of this pathway's activity during early pregnancy in vivo. For example, conditional ALK2 deletion results in female infertility due to impaired endometrial stromal cell decidualization[13], suggesting that the temporal activation of BMP signaling via ALK2 is required for the reproductive function of the uterus. Our studies presented here show abnormally elevated expression of the gene encoding the activin subunit (*Inhbb*, Supplementary Data File 4), decreased levels of its natural secreted inhibitor, follistatin, as well as disrupted expression of the BMP antagonists, Noggin (decreased) and Gremlin 2 (elevated). These findings are critical in light of the recent studies indicating that activin induces a non-signaling complex with ALK2/ACVR2A/2B, thereby inhibiting BMP signaling when local activin levels are high[62,63]. Whether impaired decidualization in our *Smad1/5* cKO and *Acvr2a* cKO models may result from the formation of an activin/ALK2 non-signaling complex remains a possibility that can be investigated in primary stromal cell cultures.

Because the BMP type 2 receptors, ACVR2A and ACVR2B, are also shared with other ligands such as activins and myostatin, which in turn activate SMAD2/3 signaling[64], it will be important to assess the interplay of these alternate pathways in the endometrium. Structural studies have shown that ACVR2A and ACVR2B are promiscuous in their binding to various TGFβ superfamily members[57,65]; therefore, the different phenotypes that we observed in *Acvr2a* cKO vs. *Acvr2b* cKO (infertility vs. subfertility) may be the result of tissue-specific abundance of BMP type 2 receptors (i.e., ACVR2A) and its BMP type 1 receptor (i.e., ALK3) during the window of implantation. This is supported by gene expression studies of *Acvr2a* and *Acvr2b* in uterine tissues, which indicate that *Acvr2a* is more abundant than *Acvr2b*.

Double conditional deletion of SMAD1/5 led to infertility, whereas single conditional deletion of SMAD1 or SMAD5 resulted in normal fertility or subfertility, respectively. These results indicate that SMAD1 and SMAD5 have both overlapping and unique functions in the uterus, and that conditional ablation of both is required to perturb the window of implantation. Redundant roles for SMAD1/5 have been previously described in the gonads, where double SMAD1/5 deletion led to metastatic

testicular and granulosa cell tumor development[66]. Molecular analysis of the endometrium of Smad1/5 cKO mice showed that the 3.5 dpc timepoint was characterized by a hyper-estrogenic status, with increased expression of E2-regulated genes (Lcn2, Ltf, Muc1), and unopposed luminal epithelial cell proliferation. At 4.5 dpc, the endometrium failed to transform into a receptive state, with cytoplasmic FOXO1 expression, luminal epithelial PR expression, and an unattached blastocyst. FOXO1 is a critical marker of endometrial receptivity that is controlled by epithelial PR; nuclear translocation of FOXO1 in the luminal epithelium is required for receptivity and embryo attachment[28]. These defects coincided with HAND2 downregulation, as well as down-regulation of other stromal PR-regulated genes such as Bmp2, Ptgs2, and Wnt4. We also observed that the IHH/COUP-TFII signaling axis was displaced and was upregulated at 4.5 dpc in the Smad1/5 cKO, likely as a reflection of the elevated epithelial PR. LIF is a cytokine that is induced by the nidatory E2 surge prior to the window of implantation[6]. Despite showing normal serum E2 levels, increased Lif was observed in the Smad1/5 cKO mice at both 3.5 and 4.5 dpc. This indicated that Smad1/5 cKO mice experienced an endometrial hyper-sensitivity to E2 during the window of implantation that combined with defective PR function, prevented the endometrium from reaching a receptive phase. Smad1/5 cKO implantation defects were not rescued by ICI 182,780 administration, indicating that combined E2 and P4 action was dysfunctional in these mice.

BMPs exert crucial roles in the glandular morphogenesis and differentiation of various tissues, including the lung, gut, mammary gland and kidney[67–69]. In the developing mouse uterus, BMP signaling via ALK6 is critical for postnatal glandular development, as ALK6 KO females lack endometrial glands[70]. Mice with conditional deletion of SMAD1/SMAD4/SMAD5 using the anti-müllerian receptor type 2-cre (Amhr2-cre), which causes recombination to occur in the endometrial stromal and uterine myometrium, also led to female fertility defects owing to abnormal decidualization[71]. These mice also developed oviductal defects, likely owing to the absence of SMAD signaling in the muscular layer of the uterus. In our study, Smad1/5 cKO mice developed enlarged cystic endometrial glands beginning at 6 weeks of age that worsened with age. 3D imaging of the chemically cleared uterus provided a comprehensive view of endometrial glands that permitted morphometric calculations regarding glandular width, length, and coiling of individual glands, indicating the genetic regulation that underlies uterine gland morphology and function. At the molecular level, the WNT/β-catenin pathway was abnormal in the endometrial tissues of Smad1/5 cKO mice, which showed upregulation of the Sfrp1-5 group of genes that antagonize WNT/β-catenin signaling[26,72]. Previous studies showed that exogenous E2 affected glandular development by increasing Sfrp2 and abrogating WNT-mediated signaling required for adenogenesis in sheep[73]. Furthermore, mice with conditional ablation of SMAD2/3 in the uterus demonstrated the relationship between TGFβ and WNT/β-catenin signaling on the development and maintenance of normal endometrial glandular structure and function[74]. Therefore, although previous evidence showed a relationship between E2 or TGFβ with the WNT/β-catenin axis, in our studies, abrogating BMP signaling in the endometrium perturbed both E2 and WNT/β-catenin signaling and led to enlarged, cystic endometrial glands. In tissues such as the breast and lung, paracrine signals from the stroma control branching morphogenesis[69]. However, because our mouse model deleted BMP signaling in both the stromal and epithelial compartments, studies using cell-specific SMAD1/5 ablation will be necessary to understand how paracrine BMP signaling affects endometrial gland development and function.

RNAseq of the 3.5 dpc pseudopregnant uterus showed that compared with controls, uteri from Smad1/5 cKO and Acvr2a cKO mice shared many differentially regulated genes. Shared cellular pathways between the two genotypes included implantation-related pathways, iron and retinoid transport and metabolism, defects in the WNT-signaling pathway, apicobasal polarity, cell–cell junction, and extracellular matrix organization. The "plasma membrane transformation" occurs prior to embryo implantation and includes loss of epithelial apicobasal polarity, loss of adherens junctions, microvilli flattening, and pinopode appearance[46,75]. RNAseq identified abnormal expression of genes involved in E2 and P4 response, such as Muc1, Inhbb, and Areg, Klf15, and Hand2. Genes involved in ciliogenesis, Nme3 and Foxj1, were also overexpressed in the mutants. Nme3 encodes a kinase that causes ciliopathy-associated phenotypes and localizes to the basal body[48]. FOXJ1 is a transcription factor that controls ciliary development[76,77]. Thus, transcriptomic profiling of Smad1/5 cKO and Acvr2a cKO uterine tissues during the window of implantation, revealed that abrogated uterine BMP signaling led to an abnormal response to nidatory E2 and defective epithelial cell remodeling during the window of implantation. Given that these data represent gene expression differences from RNAseq analyses alone, experimental validation by alternate methods would help overcome the limitations of this approach.

BMP signaling is critical for the decidualization of human endometrial stromal cells induced to decidualize in vitro[9,13]. Previous studies demonstrated that siRNA-mediated knockdown of the ALK2 receptor decreased the ability of human endometrial stromal cells to decidualize in response to E2, P4, and cyclic AMP. ALK2-mediated decidualization was shown to be driven by the downstream transcriptional activation of the transcription factor CEBPβ, a critical regulator of PR activation during decidualization[13]. Therefore, in the human endometrium, BMP/SMAD signaling is controlled via the BMP type 1 receptor ALK2. However, the ligands, type 2 receptors, and downstream activated genes during this process remain to be uncovered. In our study, we show that pSMAD1/5 is localized in the endometrial glands during the proliferative phase of the endometrium and shifts to the stroma as the endometrium transitions to the mid-secretory phase. Future studies will be necessary to outline the patterns of pSMAD1/5 expression in the late-secretory phase of the human endometrium and in true decidual tissues. Given that BMP7 mutations have been identified in patients with recurrent pregnancy loss[20], revealing the pathways regulated downstream of BMPs would be open new venues for the treatment or diagnosis of implantation defects in women with infertility.

In conclusion, our results support the hypothesis that BMPs signal via a conserved ACVR2A/ALK3/SMAD1/5 pathway during implantation. This BMP signaling pathway is critical for integrating the endometrial response to the nidatory E2 surge and P4, which directs the endometrial remodeling required to support embryo implantation and early pregnancy.

## Methods

**Animal models, breeding schemes, and fertility analyses.** For fertility analyses, 6-week-old female mice were mated to WT males of proven fertility for 6 months and the total number of pups was quantified. Smad1[flox/flox] and Smad5[flox/flox] mice were previously generated and described[78,79]. To generate Smad1/5 cKO females, female mice carrying homozygous Smad1[flox/flox]; Smad5[flox/flox] alleles were crossed to males carrying the double Smad1[flox/flox]; Smad5[flox/flox] alleles and the progesterone receptor-cre (PR[cre/+])[23]. A similar breeding scheme was used to generate Acvr2a[flox/flox]-PR[cre/+] and Acvr2b[flox/flox]-PR[cre/+] mice using Acvr2a[flox/flox] and Acvr2b[flox/flox] mice[80,81]. Mice were genotyped by PCR using genomic tail DNA with primers listed in Supplementary Table 2. The mice were maintained on a hybrid C57BL/6 J and 129S5/SvEvBrd genetic background. Animal handling and experimental studies were performed following the NIH Guide for the Care and Use of Laboratory Animals and were approved by the Institutional Animal Care and Use Committee of Baylor College of Medicine. Animals were maintained on a

12-hour light/dark cycle and in a vivarium with a controlled ambient temperature of 70°F ± 2°F and 30–70% relative humidity.

**Rodent surgeries.** Approval from the Institutional Animal Care and Use Committee at Baylor College of Medicine was obtained for these studies and their guidelines were followed on all procedures. Prior to ovariectomy, mice were injected with slow-release buprenorphine (ZooPharm) (1 mg/kg) and meloxicam (Norbrook) (4 mg/kg) to control pain and were anesthetized with 2% isoflurane (Piramal) with oxygen. We first made a 0.3–0.5 cm midline incision into the skin, then gently pulled the fat pad out of the incision to exposed the oviduct, which was then tied with absorbable Vicryl (Ethicon) followed by cutting of the ovary with small sharp scissors. We then sutured the abdomen with absorbable Vicryl, closed the skin with a surgical clip, allowed the mice to wake on a warm plate. The mice are monitored daily and injected daily with analgesics for a minimum of 72 h. For embryo transfer procedures recipient WT or cKO mice were induced with anesthesia on day 2.5–3.5 dpc of pseudopregnancy as described above, then the oviduct was exposed to allow for the transfer of WT 3.5 dpc blastocysts into the uterus that was harvested from WT or mutant donor female mice.

**Tissue acquisition, and handling for histology, nucleic-acid analyses, and qPCR.** Human endometrial tissues were obtained by obtaining informed consent from women and using IRB-approved protocols by Baylor College of Medicine and University of North Carolina School of Medicine (H-46538, H-211138). Tissues were harvested and immediately placed in 10% formalin overnight then changed to 70% ethanol (EtOH) until processing for paraffin embedding. Tissues were processed for paraffin embedding in the Pathology and Histology Core Facility at Baylor College of Medicine. Implantation sites were visualized at 4.5 dpc after mating control or mutant females to WT males and administered with Chicago Sky Blue Dye via retro-orbital injection at 4.5 dpc[82]. For RNA or protein extractions, uterine tissues were collected and immediately frozen in dry ice. RNA was extracted with RNEasy kit following the manufacturer's methods, and 1 µg was used for reverse transcription with qSCRIPT cDNA SuperMix (Quanta). cDNA was diluted threefold and 1 µl was used for qPCR using SYBR Green reagent on a Roche 480 Light Cycler II. The qPCR data were analyzed using the $^{\Delta\Delta}$Ct method[83] and analyzed with a Student's $t$ test or analysis of variance (ANOVA) with Tukey's multiple comparison post-tests on Excel or Prism GraphPad version 8. Data are plotted as mean ± standard error of the mean (SEM), *$p < 0.05$, **$p < 0.001$, ***$P < 0.0001$, or as indicated in the corresponding figure legends. Primer sequences are listed in Supplementary Table 2.

**RNA sequencing and analysis.** Uterine tissues were collected from control, Smad1/5 cKO, and Acvr2a cKO mice at 3.5 dpc of pseudopregnancy and immediately snap-frozen on dry ice or fixed in formalin. Mice were determined to be 3.5 dpc pseudopregnant according to the levels of serum P4 (~9.2–29.3 ng/µl). RNA was extracted from the tissues as described above, but processed in Trizol and isolated with the Direct-zol RNA extraction kit (Zymo). Quality control of the samples was determined by assessing the RNA integrity number and then used for library preparation and sequencing. Sequencing (>20 M reads per sample) was performed by Novogene Corporation (Santa Cruz, CA), on the Illumina Platform (PE150). Sequences were aligned and transcript abundance was performed using HISAT2 2.1.0 and cufflinks 2.2.1.2, and FPKM values were log2-transformed prior to analysis[84]. Genes with $p < 0.01$ by $t$ test and fold change >1.4 were used for supervised clustering by centering each value on the average of the control group. Gene ontology enrichment analysis was performed on all the up- and down-regulated genes using the Sigterms v1.0 program[85], and $p$ values are one-sided Fisher's exact test. Further analysis and determination of the redundancy of duplicated terms were performed with the REVIGO program v1.0 using an allowed similarity score of small (0.5)[55]. RNA-sequencing data are available in the GEO database under accession number GSE152675.

**Scanning electron microscopy.** To analyze the surface of the luminal uterine epithelium, uteri were harvested from mice at 3.5 dpc and flushed with 0.5 ml of PBS to remove blastocysts and debris. The uteri were then dissected longitudinally under a dissecting microscope with spring scissors and immediately submerged in ice-cold 2.5% glutaraldehyde in PBS overnight at 4 °C. The tissues were washed extensively in PBS and gradually dehydrated in a series of EtOH solutions for 15 min incubation times and stored in 100% EtOH. The tissues were then dried in an Autosamdri-815 (Tousimis Research Corporation) critical point dryer for 1.5 h. To enhance the SEM image contrast, the samples were coated with a thin iridium film of 7 nm with a magnetron sputtering coater (208HR High Resolution Sputter Coater, Ted Pella, Inc). Images of the uterus were collected in a Nova Nano SEM (FEI) with a working distance of 5 mm at room temperature in a high vacuum (2E$^{-6}$ Torr) at the Houston Methodist Hospital TEM Microscopy Core.

**Isolation of uterine epithelium from stroma and myometrium.** Uterine epithelial and stromal isolation was performed by enzymatic and mechanical digestion in trypsin from bovine pancreas (Sigma, T1426) dissolved in Hanks' Balanced Salt Solution (HBSS) (Gibco) under a dissecting stereomicroscope[13]. Uteri were collected and cut into 2–5 mm cross-sections, followed by incubation in 1% trypsin

solution in HBSS at 37 °C for 30–60 min. The epithelial cells were mechanically separated from the uterus using forceps to push out the epithelial sheets and a mouth pipette to capture the epithelium after dissection. The purity of the epithelial cells was determined by analyzing cytokeratin 8 (Krt8) and vimentin (Vim) mRNA by qPCR.

**Hormone treatments and ICI 182, 780 administration.** Superovulation studies were performed to assess ovarian function independently of uterine function in the Smad1/5 cKO, Acvr2a cKO, and Acvr2b cKO mice. To induce superovulation in mice, 3-week-old females were injected with five IU of pregnant mare serum gonadotropin (PMSG or equine gonadotropin). Forty-four to 46-hours later, the mice were injected with five IU of hCG. In all, 18–24 h later, the ovaries and oviducts were collected in M2 Medium (Sigma) with 0.1% hyaluronidase (Sigma), and the total number of ovulated oocytes was quantified using a stereomicroscope. To assess pre-ovulatory follicles of Acvr2a cKOs, mice were treated as above with PMSG, and ovaries were collected and formalin-fixed 6 h after hCG administration. To monitor blastocyst development WT and Acvr2a cKO mice were superovulated with PMSG and hCG and mated to WT males immediately after hCG administration. Fertilized eggs were collected in the morning after mating (0.5 dpc) and placed in M2 medium in a humidified incubator set at 37 °C. Progression of the fertilized eggs to the two-cell and blastocyst stage was assessed daily. Mice were subjected to artificial pregnancy by timed hormone injections[41] as follows: 6-week-old mice were ovariectomized, and at 8 weeks of age, they were primed with 100 ng E2 (in sesame oil), rested for 2 days, then administered with the following hormone regimen for 4 days: injected with 1 mg P4 on days 1–3, then with 1 mg P4 + 50 ng E2 on day 4. In all, 15 h after the last injection, mice were killed and RNA was isolated from whole uteri, fixed in formalin, or processed for epithelial/stromal cell isolation.

For the ICI 182, 780 experiments, 6–8 week-old control of Smad1/5 cKO mice were mated to WT males. On the morning of 3.5 dpc (0900hrs), control and Smad1/5 cKO received either injection (s.c.) of vehicle (sesame oil) or 10 ng ICI 182, 780 (Tocris, 1047)[86,87]. Because implantation in the mouse occurs at 4.5 dpc, implantation was assessed in the afternoon of 5.5 dpc using an injection of Sky Blue Dye (Sigma, C8679).

*Artificial induction of decidualization.* To test the uterine response to an artificial decidual stimulus[30], mice were ovariectomized, allowed to recover for 2 weeks, and injected subcutaneously with 100 ng E2 for 3 days. Following 2 days of rest, mice were injected with three daily subcutaneous injections of 1 mg P4 plus 6.7 ng E2 administered at 0900hrs. On the afternoon of the third injection of 1 mg P4 plus 6.7 ng E2, mice were subjected to a second surgery, where one uterine horn was exposed and injected with 50 µl of sesame oil. The contralateral horn was not injected and served as a negative control. Hormone injections (1 mg P4 plus 6.7 ng E2) continued for 5 days after the oil injection, at which point the mice were euthanized and their reproductive tracts were collected, imaged, and weighed. Tissues were frozen for mRNA expression or fixed in formalin for histological analyses.

*Antibody immunostaining.* Formalin-fixed paraffin-embedded sections were deparaffinized with Histoclear (National Diagnostics) and rehydrated in a series of EtOH solutions. Antigen retrieval was performed in a microwave for 20 min in a 10 mM Citrate, 0.05% Tween-20, pH 6.0 solution. Sections were blocked in 3% bovine serum albumin (BSA) (Sigma) for 1 h, followed by overnight incubation in the primary antibody diluted in 3% BSA (Sigma, A2153). For immunofluorescence, detection and labeling were performed with secondary antibodies conjugated to Alexa-Fluor-488 or Alexa-Fluor-594 fluorophores (Invitrogen, A21203, A21206, or A21209) at a dilution of 1:250 in 3% BSA. For peroxidase staining, sections were washed and incubated in Biotinylated secondary antibodies (Goat Anti-Rabbit IgG Biotinylated, Vector Biolabs, BA-1000 at 1:200 dilution in 3% BSA; Goat Anti-Rat IgG Biotinylated, Vector Biolabs, BA-9400, 1:200 dilution in 3% BSA; Vector Mouse on Mouse Immunodetection Kit, Vector Biolabs, FMK-220 following manufacturer's conditions), followed by avidin/biotin complex formation (Vectastain ABC, PK-6100, Vector Biolabs). Sections were then incubated with DAB peroxidase (horseradish peroxidase) substrate (Vector labs, SK-4100) and counterstained with hematoxylin. Immunofluorescence images were obtained using a Zeiss LSM 880 confocal microscope. All antibodies and dilutions are listed in Supplementary Table 3.

*Whole-mount immunostaining and 3D imaging of the uterus using OPT and multiphoton imaging.* Tissues were dissected and fixed in ice-cold 4% paraformaldehyde at 4 °C overnight. The tissues were then dehydrated in a stepwise manner with serial dilutions of EtOH mixed with PBS, then quenched in a solution of EtOH/DMSO/H$_2$O$_2$ (2:1:3, i.e.,15% H$_2$O$_2$) for 24 h at room temperature (modified from ref. [88]). Tissues were permeabilized using 3–5 freeze at −80 °C and thaw cycles in absolute EtOH, then rehydrated with stepwise dilutions of PBS-triton/EtOH to PBS-triton (0.1% triton). After blocking in 10% fetal bovine serum (FBS) in PBS-triton for 12–24 h at room temperature, the tissues were incubated in primary antibody solution containing 5% FBS in PBS-triton plus FOXA2 or E-cadherin antibody (Supplementary Table 3) for 48 h at room temperature. Excess primary antibody was washed with PBS-triton for 24 h, then incubated with

secondary antibody solution for 48 h, and tissues were then washed extensively in PBS-triton for 24 h. Specimens for OPT were prepared as described earlier[89]; in brief, uteri were embedded in 1% agarose, dehydrated in serial dilutions of PBS/EtOH for 12 h, and cleared in Benzyl alcohol/Benzyl benzoate (1:2) solution for at least 48 h. Imaging was performed on a custom-built OPT microscope at the Optical Imaging and Vital Microscopy Core Laboratory at Baylor College of Medicine. Images were reconstructed using nRecon (Skyscan Pty Ltd., Kontich, Belgium) software 1.7.1.0. Specimens for multiphoton microscopy were immunostained as described above and after PBS-triton wash cleared in scaleCUBIC-1 reagent[90]: 25% urea, 118 25 wt% N,N,N0,N0 -tetrakis(2-hydroxypropyl) ethylenediamine and 15 wt% Triton X-100 for at least 96 h. Imaging was performed on Nikon A1R MP + Multiphoton microscope in Biocenter Oulu Tissue Imaging Center at Oulu University, Finland. Analysis of both data sets was done by Imaris (Bitplane AG) software v9.2.1.

*Examination of estrous cycles and hormonal analyses.* Female mice 8 weeks of age ($n = 4–5$ per genotype) were individually housed for the duration of the study. Vaginal smears were obtained by flushing the vaginal opening with 20 μl of PBS each morning between 9–10 am. Vaginal smears were placed inside the well of a 24-well plate and examined under a microscope for the presence of leukocytes, cornified or nucleated epithelium; stages of the estrous cycle were assigned according to previously described criteria, and each cycle defined as the sequential completion of proestrus, estrus and diestrus/metestrus[91]. Hormone assays were performed by the University of Virginia Center for Research in Reproduction Ligand Assay and Analysis Core (NICHD grant R24 HS102061). Statistical analyses were performed by a one-way ANOVA, followed by a Tukey's multiple comparison post hoc test on GraphPad Prism 8.

*Statistics and reproducibility.* Statistical analyses were performed on GraphPad Prism 8 or Microsoft Excel 16.47.1. Statistical tests, including *p* values, are reported in the corresponding figure legends or, when possible, directly on the data image. To ensure the reproducibility of our findings, experiments were replicated in a minimum of three independent samples, to ensure biological significance, and at least three independent times to ensure technical and experimental rigor and reproducibility.

**Reporting summary**. Further information on research design is available in the Nature Research Reporting Summary linked to this article.

## Data availability

The authors declare that all data supporting the findings of this study are available within the article and its supplementary information files or from the corresponding author upon reasonable request. The data sets generated in this study have been deposited in the Gene Expression Omnibus database under Accession Code: GSE152675.

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

## Acknowledgements

We thank Dr. Yasmin M. Vasquez for the critical reading of the manuscript. Studies were supported by *Eunice Kennedy Shriver* National Institute of Child Health and Human Development grants K99/R00-HD096057 (to D.M.) and R01-HD032067 (to M.M.M.), NIH grant R01-AR060636 (to S.L.), NIH grant CA125123 (to C.J.C.) and the National Institute of Environmental Health Sciences (to F.J.D.). Diana Monsivais, Ph.D. holds a Postdoctoral Enrichment Program Award from the Burroughs Wellcome Fund.

## Author contributions

Study conception and design: D.M., T.N., R.P.H., K.N., K.S., F.J.D., M.I., S.J.L., M.M.M. Performed experiment or data collection: D.M., T.N., R.P.H., K.N., K.S., C.H., J.A., S.T. Computation and statistical analysis: F.C., C.J.C. Data interpretation and analysis: D.M., T.N., R.P.H., K.N., K.S., F.J.D., M.I., S.J.L., S.L.Y., R.M., M.M.M. Writing, reviewing and editing: All. Supervision: D.M., M.M.M.

## Competing interests

The authors declare no competing interests.
