## [Peer Review File · Nature Communications]

Reviewers' Comments:

Reviewer #1:

Remarks to the Author:

During early pregnancy in the mouse, nidatory estrogen (E2) stimulates endometrial receptivity by activating a network of signaling pathways that is not yet fully characterized. In this study, the investigators found that BMPs regulate the endometrial response to nidatory E2 via a conserved activin receptor type 2A (ACVR2A) and SMAD1/5 signaling pathway. Mice with single or double conditional deletion of SMAD1/5 and ACVR2A/ACVR2B receptors using a Pgr-Cre showed that SMAD1/5 deletion showed enhanced E2 response that resulted in the development of cystic endometrial glands, a hyperproliferative during the window of implantation, and derailed apicobasal transformation that prevented embryo implantation and infertility. Analysis of *Acvr2a-PRcre* and *Acvr2b-PRcre* pregnant mice determined that BMP signaling occurred via ACVR2A; ACVR2B was dispensable during embryo implantation. Therefore, BMPs signal through a conserved endometrial ACVR2A/SMAD1/5 pathway that integrates the nidatory E2 response during embryo implantation.

This is an interesting straightforward, but another knockout study, showing exaggerated estrogenic responses in SMAD1/5 mutant mice that disrupts implantation. The adverse effects of SMAD1/5 have previously been shown by the same authors, raising concerns for the novelty of this study. Moreover, there are several concerns with this study that are described below.

1. The investigators should measure E2 and P4 levels during the window of implantation to show E2 surge and also define window of implantation in WT and mutant females with respect to time and changes in marker genes.
2. Can the hyper-estrogenic milieu be overcome by the administration of excess P4? What are the criteria for assaying enhanced E2 responses?
3. Can the enhanced E2 responses be subdued by an injection of the estrogen receptor antagonist ICI 182,780?
4. Do these mutant mice become pseudopregnant in the absence of embryo?
5. What is the phenotype of ACVR2A mutant mice? What is status of SMADs in the absence of ACVR2A?
6. LEFTY1 is known to suppress SMAD2 and SMAD5 activation in basal cells and breast cancer cells (Cell Stem Cell, August 2020). What is the status of Lefty1 in the uterus during implantation in mutant females?
7. The quality of several images should need improvement and the figures should be self-explanatory.
8. It is prudent to show the spatiotemporal expression SMAD1/SMAD5 and of ACVR2A before and during implantation.
9. What are the status of ovarian activity (follicle and Corpus luteum) in the mutant mice, since SMAD1/SMAD5 and ACVR2A are likely to be expressed in the ovary.

Reviewer #2:

Remarks to the Author:

Overview

This paper investigates the contribution of SMAD1/5 and *Acvr2a*/*Acvr2b* to implantation in mice. Understanding the processes involved in implantation is critical to improve treatments for those with early pregnancy problems such as recurrent miscarriage and implantation failure.

The manuscript describes the localisation of pSMAD1/5 in wildtype mice during the early pregnancy (1.5-4.5dpc). Conditional KO of *Smad 1* or *Smad 5* revealed subfertility in *Smad5* cKO

mice due to haemorrhage and resorption at implantation sites. Mice with double conditional knockout of Smad1/5 were infertile. Non-pregnant Smad1/5 cKO mice had abnormal glands and a reduced response to artificial decidualisation. Pregnant Smad1/5 cKO mice had abnormal glands (4.5dpc), abnormal proliferation (3.5dpc) and no implantation sites (4.5dpc).

Acvr2a cKO mice were infertile, Acvr2b cKO mice were subfertile. Acvr2b cKO mice had no difference in number or weight of implantation sites at 4.5dpc and artificial decidualisation was not altered. There was evidence of implantation site resorption at 10.5dpc and absence of uNK cells. Acvr2a cKO mice had no implantation sites. They had normal estrous cycles but did not respond to artificial decidualisation. Endometrial defects were confirmed by lack of implantation sites with embryo transfer of wild type embryos into Acvr2a cKO recipients.

RNASeq analysis revealed shared signalling pathways between Smad1/5 cKO and Acvr2a cKO mice. Scanning electron microscopy showed lack of pinopods and dense microvilli, consistent with reduced remodelling of the luminal epithelium.

This is a well written paper in an important area and the methods used are sound. I have only a few general and specific comments/suggestions with the aim of improving the manuscript as I feel it is suitable for publication in Nature Communications.

General comments

1. Important scientific data is included in the supplementary figures. This makes it more difficult for the reader to follow the scientific logic of the paper. I would suggest moving the Smad1 and Smad 5 cKO data into the figure 1. The graphs of pups/female could be presented as in Fig 4B and then Supplementary Figure 1 merged into Figure 1. The current Figure 1 I-M could be moved to Supplementary Figure 1. Suppl fig 3D-0 may be better placed as part of figure 2. Suppl figure 4I-M could be part of Figure 3.
2. Transfer of Smad1/5 cKO embryos to WT recipients and WT embryos to Smad1/5 cKO recipients would be the definitive experiment to confirm an endometrial defect in these mice. This was performed for Acvr2a cKO but not Smad1/5 cKO. Given that TGFB superfamily growth factors can influence oocyte quality and maternal-embryo communication, it would be great to have this data included.
3. It would be of interest to show the implantation rates when Acvr2a cKO embryos were transferred to wild type recipient females to determine if there is any embryo contribution to the infertility seen in this cKO.
4. These data are very interesting but entirely derived from mice. There is minimal mention of the clinical application of the work. It would be of interest to see pSMAD1/5 IHC in human endometrium during the window of implantation, alongside the mouse tissue staining. At a minimum, the discussion should address the translation of these important findings in mouse tissue to humans, e.g. findings in humans to date, clinical context, any limitations, suggested future work with human tissue.

Specific comments

1. Table 1: please make clear in the title that this table illustrates only selected differentially regulated genes, i.e. those known to have a role in endometrial receptivity.
2. Figure 6 legend: please define arrows in F, H and J.
3. A previous study of SMAD 1/5/4 signalling during early pregnancy in the mouse is not discussed. DOI: 10.1095/biolreprod.116.139477

Reviewer #3:

Remarks to the Author:

This study by Matzuk and colleagues highlights a new mechanism, by which the endometrium responds to nidatory estrogen (E2) to prepare for successful embryo implantation. The group describes BMPs and in particular the BMP receptor ACVR2A and SMAD1/5 to be the upstream regulators for endometrial receptivity as conditional knockout mouse models (for ACVR2 receptors and SMAD1 and SMAD5) show overlapping effects and displayed enhanced E2 response, development of cystic endometrial glands, hyperproliferation and aberrant apicobasal transformation of the endometrial epithelium, all preventing embryo implantation.

This article is very interesting and of high relevance for numerous reasons:

- Thoughtful selection of mouse models, that cover all necessary controls to come up with the relevance of ACVR2A/ALK3/Smad1/5 in preparing the endometrium for embryo implantation via a strict temporal regulation of E2 expression in early steps of pregnancy.
- The in vivo work is supported by RNA Seq data to unravel the molecular processes relevant for these findings, e.g. regulation of genes affecting the apicobasal polarity of the epithelial cells and others.
- quality of the figures is high and techniques chosen sufficient to approve the statements

However there are some concerns/points that need to be addressed for clarification:

1. line 113-115 refers to the study on ALK2 and BMP2. BMP2 is not the right ligand to this receptor, therefore the argumentation there is misleading and confusing. Also, later in the discussion ALK2 is not mentioned again. ALK2 forms with ACVR2A/B a non-signaling complex, also when bound to Activin A, as shown by several groups now. Which role might this play in the study presented here? This might be interesting also to my point 4 below.
2. Fig 2: Fig 2 M to O' are better to see than Fig I to L; therefore M to O' should be enlarged at the expense of I to L.
3. Fig 3P: what about BMPs other than BMP2 and Activin? what about frizzled receptors?
4. Table 1, line 346: refer to noggin in this context- also why are both follistatin and noggin affected with follistatin being an activin and noggin a BMP antagonist. This might be interesting and should be addressed here.
5. excellent and sufficient discussion on the ligands (hinting towards the potential contribution of hetero-dimers) and receptors; only ALK2 needs to be considered also here, especially as ref 13 (Clementi et al) is mentioned here as well.
6. lines 432-433: where the same genes affected in the ACVR2B cKO? Please discuss this.

RESPONSE TO REVIEWER'S COMMENTS

We would like to apologize for the delayed response to the reviewers' comments; however, setbacks due to COVID19 restrictions drastically slowed down the breeding of our mouse colonies and in some cases, limited personnel due to illness. We truly appreciate the reviewers' thorough review of our manuscript and have strongly considered each of their comments and suggestions. The revised manuscript now includes additional experimental figures, numerous revisions to the text, and reformatting of the figures as suggested by the reviewers. The new suggested experiments improved the quality of our manuscript for publication in *Nature Communications* and supported our conclusions.

Our detailed responses are indicated below by blue font, while the editor's and reviewer's comments are in black.

Dear Dr Monsivais,

I apologise for the delay in contacting you with this editorial decision on your manuscript.

Thank you again for submitting your manuscript "Endometrial receptivity and implantation require uterine BMP signaling through an ACVR2A-SMAD1/SMAD5 axis" to *Nature Communications*. We have now received reports from three reviewers and, after careful consideration, we have decided to invite a major revision of the manuscript.

As you will see from the reports copied below, the reviewers raise important concerns. We find that these concerns limit the strength of the study, and therefore we ask you to address them with additional work. Without substantial revisions, we will be unlikely to send the paper back to review. In particular, please address the various points of all reviewers to clarify the effects of various hormones (see comments from Referee #1) and to include the mutants suggested by Referee #2 to confirm the endometrial defects. As I am sure you are aware, irrespective of whether new data are required to address a reviewer query, we would expect you to address all points in detail in your response to the reviewers/a revised manuscript (if appropriate).

If you feel that you are able to comprehensively address the reviewers' concerns, please provide a point-by-point response to these comments along with your revision. Please show all changes in the manuscript text file with track changes or colour highlighting. If you are unable to address specific reviewer requests or find any points invalid, please explain why in the point-by-point response.

Reviewer #1 (Remarks to the Author):

1. The investigators should measure E2 and P4 levels during the window of implantation to show E2 surge and also define window of implantation in WT and mutant females with respect to time and changes in marker genes. We appreciate this reviewer's insightful comment. To address this question, we have performed analyses of E2 and P4 serum levels from mice subjected to timed pregnancy by mating to vasectomized males to induce pseudopregnancy. We first confirmed the state of pseudopregnancy in the Control and mutant lines by the rise in P4 levels in mice relative to a non-pregnancy state (New Supplemental Figure S8A). At this timepoint, we also observed a slight increase in serum E2 levels in the control, *Smad1/5* cKO and *Acvr2a* cKO mice relative to non-pregnant Control mice during the diestrus phase (New Supplemental Figure S8B). Although the increase was not statistically different to the serum E2 levels observed in the non-pregnant diestrus phase, it is consistent with the previously reported changes in the literature (PMID 4452972, 5459217). Because of variable mating and ovulation schedules, we found it difficult to capture the sensitive serum estradiol changes during the window of implantation. Future studies will be necessary to carefully outline this in natural pregnancy in the mouse.

To define the window of implantation in control and mutant mice, we performed analysis of the endometrium at 3.5dpc of pseudopregnancy in the group of mice described above. We performed IHC to determine the status of luminal epithelial cell proliferation (Ki67) (New Supplemental Figure S8C-E) and analyzed the expression of genes involved in endometrial receptivity by RNAseq (New Supplemental Figure S8J-U). A receptive endometrium is characterized by reduced proliferation of the luminal epithelium, which we detected in the endometrium of Control mice at 3.5 dpc, but not in the endometrium of *Smad1/5* cKO or *Acvr2a* cKO mice, which showed unopposed epithelial cell proliferation (Figure S8C-E). We found that the expression of several genes known to be critical for receptivity were significantly downregulated in the uterine tissues of the *Smad1/5* cKO and *Acvr2a* cKO mice (i.e., *Fst*, *Hand2*, *Nr2f2/Coup-TFII*, *Ptch1*, *Egfr*, *Klf15*, and *Areg*). We also observed that the mutant mouse lines demonstrated elevated expression of genes that are typically downregulated during the window of implantation (i.e., *Muc1*, *Inhbb/Activin*, *Crabp2*, *Clca1*, and *Lcn2*). Thus, by performing gene expression and proliferation marker analyses, we identified that the window of implantation is abnormal in the *Smad1/5* cKO and *Acvr2a* cKO female mice. These results are included as a new supplemental figure Figure S8 and are presented in the results sections, Lines 373-382 and 397-404.

2. Can the hyper-estrogenic milieu be overcome by the administration of excess P4? What are the criteria for assaying enhanced E2 responses?

This is a great question given the critical role of P4 in the establishment and progression of pregnancy. To address this question, we tested the endometrial response to E2 and P4 in ovariectomized mice (new Supplemental Fig. 2). To determine whether the endometrial response to E2 could be suppressed by P4 administration, we quantified the expression of E2-regulated genes (*Clca2*, *Lcn2*, *Ltf*), as well as the expression of the genes encoding the estrogen and progesterone receptors (*Esr1* and *Pgr*) in the endometrial epithelium of the treated mice. We also assessed luminal uterine epithelial proliferation by IHC staining with the proliferation marker, Ki67 (new Supplemental Fig. 2C-F). We observed that the hyper-estrogenic endometrial response, as assessed by the expression of E2-regulated genes (*Clca2*, *Lcn2*, *Ltf*), and luminal epithelial proliferation (assessed by Ki67 IHC), was not overcome by P4 administration. These new data have been included as a new supplemental figure (Supplemental Fig. 2) and are integrated into the text, Lines 213-226.

3. Can the enhanced E2 responses be subdued by an injection of the estrogen receptor antagonist ICI 182,780?

We thank the reviewer for suggesting this informative experiment to determine whether the enhanced E2 action can be subdued by the ICI 182,780 antagonist. We performed this experiment in Control and *Smad1/5* cKO mice and found that ICI administration was not able to rescue the implantation defect in the mutant mice. These results suggest that perturbed implantation in the *Smad1/5* cKO mice is multifactorial and not only a result of unopposed E2 action during implantation. This is supported by our observations of the abnormal localization of PR at 3.5dpc (Fig. 3N-O), indicating that both E2 and P4 action are perturbed in these mice. These new experiments are now included in a new Supplemental Figure 3 and referenced in the text, Lines 246-255.

4. Do these mutant mice become pseudopregnant in the absence of embryo?

We appreciate this reviewer's question. To determine whether the mice became pseudopregnant, we mated control and mutant females to vasectomized males and analyzed serum levels of P4. The levels of serum P4 at 3.5 days of pseudopregnancy indicated that all three mouse lines achieved elevated serum P4 relative to non-pregnant mice (New Supplemental Fig. 8A). The uterus of the mutant mice, however, failed to decidualize in the absence of an embryo as can be seen in Fig. 3 (Q-U) and Figure 5 (I-M). Therefore, we conclude that the ovarian response to pseudopregnancy was not affected in the mutant mice. However, the uterine response to artificial decidualization was affected in the mutant mice.

5. What is the phenotype of ACVR2A mutant mice? What is status of SMADs in the absence of ACVR2A?

The reviewer raises two important questions. *Acvr2a* cKO female mice were sterile and did not generate any pups over the course of a 6-month fertility trial (Supplemental Table 1: Control, n=10, 56.4 ± 12.74 pups/female vs. *Acvr2a* cKO, n=8, 0 ± 0 pups/female). To determine the status of the SMADs in the absence of ACVR2A, we performed gene expression analysis of the genes encoding SMAD1 and SMAD5 (*Smad1* and *Smad5*) in the uterus

of control, *Smad1/5* cKO, and *Acvr2a* cKO mice and found no significant difference in the expression *Smad1* or *Smad5* in the uterine tissues of *Acvr2a* cKO mice (New Supplemental Figure 8G-H). Interestingly, we did find that *Acvr2a* levels increased significantly in the uterine tissues of *Smad1/5* cKO mice, indicating a potential mechanism of compensation (New Supplemental Figure 8F). These new studies are now included as a new supplemental figure (Supplemental Figure 8F-H) and have been incorporated into the text, lines 382-386.

6. LEFTY1 is known to suppress SMAD2 and SMAD5 activation in basal cells and breast cancer cells (Cell Stem Cell, August 2020). What is the status of Lefty1 in the uterus during implantation in mutant females?

This is a great question regarding the critical role of LEFTY1 in the endometrium and in breast cancer cells. To determine the status of LEFTY1 in the uterus during implantation, we performed qPCR analysis of *Lefty1* in uterine tissues of Control, *Smad1/5* cKO, and *Acvr2a* cKO mice collected at 3.5dpc of pseudopregnancy (New Supplemental Figure S8I). We found that the expression of *Lefty1* was unchanged across all three genotypes, indicating that in the endometrium, deletion of SMAD1/5 or ACVR2A had no effect on *Lefty1* gene expression at this timepoint. This is now incorporated into the text, lines 386-388.

7. The quality of several images should needs improvement and the figures should be self-explanatory.

We thank the reviewer for this suggestion. We have re-labeled and re-organized many of our figures throughout the manuscript (in particular, Figures 1, 2, and 3) and anticipate that the reviewers will find that the figures have improved in quality and are self-explanatory to the readers.

8. It is prudent to show the spatiotemporal expression SMAD1/SMAD5 and of ACVR2A before and during implantation.

The reviewer raises a critical point that we have addressed in the revised manuscript. We have included a time-course analysis of the active form of SMAD1/SMAD5 (phosphorylated SMAD1/5) during early pregnancy in the mouse and in human endometrial biopsies collected from the proliferative and mid-secretory phases of the cycle (Fig 1 A-H, I-N). The mouse and human IHC shows that pSMAD1/5 levels are dynamically expressed. As can be observed in Figure 1A-H, pSMAD1/5 is detected in both epithelial and stromal compartments of the endometrium at 1.5dpc (Fig. 1A-B) and 2.5dpc (Fig. 1C-D). We then observed a transient decrease in the luminal epithelium at 3.5dpc (Fig. 1E-F). At 4.5dpc we observed an interesting pattern in pSMAD1/5 within the implantation chamber of the mouse: pSMAD1/5 was readily detected in the epithelial and stromal compartments encapsulating the mouse, with restricted expression in the stroma that excluded the primary decidual zone (dotted black line, Fig. 1G-H). We observed that the human endometrium had strong glandular staining during the proliferative phase, while strong decidual cell staining was detected in the mid-secretory phase endometrium (Fig. 1I-N). Likewise, compared to the non-pregnant uterus, we observed that ACVR2A is strongly expressed in the stroma and in the apical region of the luminal uterine epithelium during implantation at 4.5dpc (Fig. 5A-F). These results are incorporated into the manuscript as Fig. 1 and Fig. 5, and in the text (lines 136-147 and lines 316-318).

9. What are the status of ovarian activity (follicle and Corpus luteum) in the mutant mice, since SMAD1/SMAD5 and ACVR2A are likely to be expressed in the ovary.

The reviewer raises an important point, which we have addressed by performing histological analyses of ovaries from control and mutant mice. Ovaries obtained from randomly cycling Control and *Smad1/5* cKO mice demonstrated the presence of several corpora lutea (CL) (Supplemental Figure S1 K-L). Likewise, we detected the presence of several CLs in the ovaries from control and *Acvr2a* cKO mice that were induced to ovulate by administration of PMSG and hCG (Supplemental Fig. 5F-G). Therefore, despite expression in the ovary, deletion of SMAD1/SMAD5 and ACVR2A does not affect CL formation.

Reviewer #2 (Remarks to the Author):

Overview

This paper investigates the contribution of SMAD1/5 and *Acvr2a*/*Acvr2b* to implantation in mice. Understanding the processes involved in implantation is critical to improve treatments for those with early pregnancy problems

such as recurrent miscarriage and implantation failure.

The manuscript describes the localisation of pSMAD1/5 in wildtype mice during the early pregnancy (1.5-4.5dpc). Conditional KO of Smad 1 or Smad 5 revealed subfertility in Smad5 cKO mice due to haemorrhage and resorption at implantation sites. Mice with double conditional knockout of Smad1/5 were infertile. Non-pregnant Smad1/5 cKO mice had abnormal glands and a reduced response to artificial decidualisation. Pregnant Smad1/5 cKO mice had abnormal glands (4.5dpc), abnormal proliferation (3.5dpc) and no implantation sites (4.5dpc).

Acvr2a cKO mice were infertile, Acvr2b cKO mice were subfertile. Acvr2b cKO mice had no difference in number or weight of implantation sites at 4.5dpc and artificial decidualisation was not altered. There was evidence of implantation site resorption at 10.5dpc and absence of uNK cells. Acvr2a cKO mice had no implantation sites. They had normal estrous cycles but did not respond to artificial decidualisation. Endometrial defects were confirmed by lack of implantation sites with embryo transfer of wild type embryos into Acvr2a cKO recipients.

RNASeq analysis revealed shared signalling pathways between Smad1/5 cKO and Acvr2a cKO mice. Scanning electron microscopy showed lack of pinopods and dense microvilli, consistent with reduced remodelling of the luminal epithelium.

This is a well written paper in an important area and the methods used are sound. I have only a few general and specific comments/suggestions with the aim of improving the manuscript as I feel it is suitable for publication in Nature Communications.

General comments

1. Important scientific data is included in the supplementary figures. This makes it more difficult for the reader to follow the scientific logic of the paper. I would suggest moving the Smad1 and Smad 5 cKO data into the figure 1. The graphs of pups/female could be presented as in Fig 4B and then Supplementary Figure 1 merged into Figure 1. The current Figure 1 I-M could be moved to Supplementary Figure 1. Suppl fig 3D-0 may be better placed as part of figure 2. Suppl figure 4I-M could be part of Figure 3.

We are grateful for this reviewer's constructive feedback and agree with the suggestions to represent the data in a more logical manner. We have taken all of these points into consideration and rearranged the figures and text accordingly. The paper is more logically organized and readable now.

2. Transfer of Smad1/5 cKO embryos to WT recipients and WT embryos to Smad1/5 cKO recipients would be the definitive experiment to confirm an endometrial defect in these mice. This was performed for Acvr2a cKO but not Smad1/5 cKO. Given that TGF β superfamily growth factors can influence oocyte quality and maternal-embryo communication, it would be great to have this data included.

Thank you for pointing out this critical experiment. To address this important point, we have performed embryo transfers as recommended. We found that when transferring embryos derived from control donor females into Smad1/5 cKO female recipients embryos failed to implant. Conversely, when embryos derived from Smad1/5 cKO female donors were transferred to control recipients, the embryos were able to implant and were grossly visible at 5.5dpc. These results indicated that infertility in the mutant females was likely the result of failed endometrial receptivity independent of the embryonic contribution. These new results have been included as new Supplemental Figure 7, and referenced in the text, Lines 358-364.

3. It would be of interest to show the implantation rates when Acvr2a cKO embryos were transferred to wild type recipient females to determine if there is any embryo contribution to the infertility seen in this cKO.

We thank the reviewer for pointing out this important experiment. We had not included this experiment in the original version of the manuscript because we did not expect any embryonic defects in the offspring of the mutant females, given that they are mated to WT males and because progesterone receptor-cre is activate in the post-natal period. However, to rule out any potential embryonic contributions, we performed embryo transfers from control, Acvr2a cKO and Smad1/5 cKO mice to WT female recipient mice. We found that embryos derived from the Acvr2a cKO and Smad1/5 cKO mutant mice were able to implant when transferred into the uteri of pseudopregnant WT mice. We conclude that there is no embryonic contribution to the infertility observed in the

Acvr2a cKO mutant mice. These results are included in Supplemental Figure 7F-I, and referenced in the text, Lines 364-369.

4. These data are very interesting but entirely derived from mice. There is minimal mention of the clinical application of the work. It would be of interest to see pSMAD1/5 IHC in human endometrium during the window of implantation, alongside the mouse tissue staining. At a minimum, the discussion should address the translation of these important findings in mouse tissue to humans, e.g. findings in humans to date, clinical context, any limitations, suggested future work with human tissue.

The reviewer raises a critical point related to our study. To address this comment, we obtained human endometrial biopsies that were dated according to their phase in the menstrual cycle by serum LH quantification or histologically by a clinical pathologist. We then performed IHC for pSMAD1/5 in cross-sections of these biopsies and observed that pSMAD1/5 expression was localized in the glands of proliferative phase endometrium and in the stromal/decidual cells of the mid-secretory phase endometrium. These findings are now included in the Revised Figure 1I-N. We have also addressed how the studies presented here translate to human health and have expanded our Introduction and Discussion sections to include studies related to BMP signaling performed in human endometrium and infertility. These are now included in lines 121-124, 143-147, and 550-563 of the manuscript.

Specific comments

1. Table 1: please make clear in the title that this table illustrates only selected differentially regulated genes, i.e. those known to have a role in endometrial receptivity.

We apologize for our lack of clarity in this table, and we have now edited the title of Table 1 to state, *“Selected differentially expressed genes in the uterus of *Smad1/5* cKO and *Acvr2a* cKO mice at pseudopregnancy day 3.5. Genes are categorized by functional pathway.”*

2. Figure 6 legend: please define arrows in F, H and J.

We thank the reviewer for pointing this out. We have included a new description of the arrows in the Figure 6 legend. Lines 1228-1230 now state,

*“Yellow arrows in the Control uterus (F) indicate that E-cadherin immunoreactivity was decreased in the basal region of the luminal epithelium of control mice but maintained in the *Smad1/5* cKO (H) and *Acvr2a* cKO (J) mice.”*

3. A previous study of SMAD 1/5/4 signalling during early pregnancy in the mouse is not discussed. DOI: 10.1095/biolreprod.116.139477

We appreciate that the reviewer has pointed out this mistake. We have added this important reference into the text (Lines 514-518, new reference number 71).

Reviewer #3 (Remarks to the Author):

This study by Matzuk and colleagues highlights a new mechanism, by which the endometrium responds to nidatory estrogen (E2) to prepare for successful embryo implantation. The group describes BMPs and in particular the BMP receptor ACVR2A and SMAD1/5 to be the upstream regulators for endometrial receptivity as conditional knockout mouse models (for ACVR2 receptors and SMAD1 and SMAD5) show overlapping effects and displayed enhanced E2 response, development of cystic endometrial glands, hyperproliferation and aberrant apicobasal transformation of the endometrial epithelium, all preventing embryo implantation.

This article is very interesting and of high relevance for numerous reasons:

- Thoughtful selection of mouse models, that cover all necessary controls to come up with the relevance of ACVR2A/ALK3/Smad1/5 in preparing the endometrium for embryo implantation via a strict temporal regulation of E2 expression in early steps of pregnancy.

- The in vivo work is supported by RNA Seq data to unravel the molecular processes relevant for these findings, e.g. regulation of genes affecting the apicobasal polarity of the epithelial cells and others.
- quality of the figures is high and techniques chosen sufficient to approve the statements

However there are some concerns/points that need to be addressed for clarification:

1. line 113-115 refers to the study on ALK2 and BMP2. BMP2 is not the right ligand to this receptor, therefore the argumentation there is misleading and confusing. Also, later in the discussion ALK2 is not mentioned again. ALK2 forms with ACVR2A/B a non-signaling complex, also when bound to Activin A, as shown by several groups now. Which role might this play in the study presented here? This might be interesting also to my point 4 below.

We thank the reviewer for pointing out this deficiency in our description of the signaling receptors. We apologize that we did not convey the information correctly and have edited the description in lines 112-145. The description now states,

“In the uterus, in vivo studies have shown that conditional deletion of BMP2 or ALK2 results in female infertility due to defects in the post-implantation process of stromal cell decidualization.”

We have also included a discussion related to the formation of a non-signaling complex between ALK2/ACVR2A/B when bound to Activin A, as suggested by this reviewer. This new information can be found in the Discussion section, Lines 469-480:

“Additional mouse models have demonstrated the role of the BMP signaling pathway in the post-implantation period and point to the delicate balance of this pathway’s activity during early pregnancy in vivo. For example, conditional ALK2 deletion results in female infertility due to impaired endometrial stromal cell decidualization¹³, suggesting that the temporal activation of BMP signaling via ALK2 is required for the reproductive function of the uterus. Our studies presented here show abnormally elevated expression of the gene encoding the activin subunit (Inhbb, Table 1), decreased levels of its natural secreted inhibitor, follistatin, as well as disrupted expression of the BMP antagonists, Noggin (decreased) and Gremlin2 (elevated). These findings are critical in light of the recent studies indicating that activin induces a non-signaling complex with ALK2/ACVR2A/2B, thereby inhibiting BMP signaling when local activin levels are high^{57, 58}. Whether impaired decidualization in our Smad1/5 cKO and Acvr2a cKO models may result from the formation of an activin/ALK2 non-signaling complex remains a possibility that can be investigated in primary stromal cell cultures.”

2. Fig 2: Fig 2 M to O' are better to see than Fig I to L; therefore M to O' should be enlarged at the expense of I to L.

We appreciate the reviewer’s suggestion to re-organize the results for clarity. We have taken these recommendations into account and have rearranged Figure 2. The changes have improved the quality and presentation of the results.

3. Fig 3P: what about BMPs other than BMP2 and Activin? what about frizzled receptors?

We are grateful that the reviewer has raised this point. To answer this question, we have expanded our analysis of the BMPs, BMP inhibitors, additional ligands of the TGFβ signaling pathway, frizzled receptors and secreted frizzled related proteins. The data are presented in Table 1 and show that several genes in these pathways are differentially expressed in the uterine tissues of the Smad1/5 cKO and Acvr2a cKO mice.

4. Table 1, line 346: refer to noggin in this context- also why are both follistatin and noggin affected with follistatin being an activin and noggin a BMP antagonist. This might be interesting and should be addressed here.

The reviewer raises a great point regarding the roles of the secreted protein antagonists of BMP and activin. We have included this point and have revised the text to state the following, Lines 266-268:

“BMP and activin-induced signaling is controlled by secreted protein antagonists, Noggin (BMP-selective) and follistatin (activin-selective), which sequester the ligands and prevent the formation of an active signaling receptor complex.”

We also describe additional members of the BMP signaling pathway that were found to be differentially regulated by RNAseq profiling, Table 1 and Lines 405-408:

“We observed abnormal expression of gene families involved in the BMP and WNT/ β -catenin signaling pathway, such as follistatin (Fst), noggin (Nog), gremlin 2 (Grem2), inhibin beta b (Inhbb), bone morphogenetic protein 4 (Bmp4), as well as several of the genes encoding the frizzled and secreted frizzled related protein gene family (Table 1).”

5. excellent and sufficient discussion on the ligands (hinting towards the potential contribution of hetero-dimers) and receptors; only ALK2 needs to be considered also here, especially as ref 13 (Clementi et al) is mentioned here as well.

We appreciate this reviewer’s positive comments regarding our discussion section. As suggested, we have expanded the discussion to include commentary related to the role of ALK2 in this signaling pathway, Lines 469-480:

“Additional mouse models have demonstrated the role of the BMP signaling pathway in the post-implantation period and point to the delicate balance of this pathway’s activity during early pregnancy in vivo. For example, conditional ALK2 deletion results in female infertility due to impaired endometrial stromal cell decidualization¹³, suggesting that the temporal activation of BMP signaling via ALK2 is required for the reproductive function of the uterus. Our studies presented here show abnormally elevated expression of the gene encoding the activin subunit (Inhbb, Table 1), decreased levels of its natural secreted inhibitor, follistatin, as well as disrupted expression of the BMP antagonists, Noggin (decreased) and Gremlin2 (elevated). These findings are critical in light of the recent studies indicating that activin induces a non-signaling complex with ALK2/ACVR2A/2B, thereby inhibiting BMP signaling when local activin levels are high^{62, 63}. Whether impaired decidualization in our Smad1/5 cKO and Acvr2a cKO models may result from the formation of an activin/ALK2 non-signaling complex remains a possibility that can be investigated in primary stromal cell cultures.”

6. lines 432-433: where the same genes affected in the ACVR2B cKO? Please discuss this.

We thank the reviewer for raising this important point. Unfortunately, the gene expression levels E2- and P4-regulated genes were not assessed in the *Acvr2b* cKO mice. Because we observed no defects during the implantation process at 4.5 days of pregnancy, analysis of the uterine tissues was not performed for this mouse line at this specific timepoint. As suggested, this is now included in the Discussion, Lines 453-456,

*“Using mice with conditional deletion of ACVR2B, we found that signaling via this receptor was dispensable during implantation, and that even though the mice experienced subfertility, the defects arose due to defects at mid-gestation. Therefore, no further mechanistic studies were performed with the *Acvr2b* cKO mice during implantation.”*

Reviewers' Comments:

Reviewer #1:

Remarks to the Author:

The investigators responded satisfactorily. Although rationale for some experiments are not presented, such as superovulation. It is to be noted that superovulation with PMSG and HCG leads to defective embryonic development and increases incidence of resorptions in the uterus after superovulation even in WT mice (Look at old papers from David Armstrong's lab (Canada). Why the investigators did not perform any analysis of CLs if they are healthy or luteolytic? Did the ICI-treated mice show normal implantation in WT mice on day 8?

Reviewer #2:

Remarks to the Author:

The authors have extensively revised the manuscript and have fully addressed all of my concerns. The flow is more logical and human endometrium/implantation is appropriately discussed. I have one suggestion and two very minor amendment requests.

1. I would suggest placing less emphasis on the role of E2 in the abstract, given that E2 antagonism did not affect implantation in controls and did not rescue implantation in genetically altered models.

2. Line 93, please change 'both endometrial cell types' to 'Two endometrial cell types'. The endometrium is multicellular and its composition dynamic, changing across the cycle.

3. Line 112-3 'In the mouse uterus'

RESPONSE TO REVIEWERS' COMMENTS

Reviewer #1 (Remarks to the Author):

The investigators responded satisfactorily. Although rationale for some experiments are not presented, such as superovulation. It is to be noted that superovulation with PMSG and HCG leads to defective embryonic development and increases incidence of resorptions in the uterus after superovulation even in WT mice (Look at old papers from David Armstrong's lab (Canada). Why the investigators did not perform any analysis of CLs if they are healthy or luteolytic? Did the ICI-treated mice show normal implantation in WT mice on day 8?

The reviewer raises the important point that superovulation in mice leads to embryonic resorption in the uterus. We would like to clarify that the superovulation studies were performed solely to determine whether a corpus luteum (CL) formed in *Acvr2a* cKO mice and were independent of our studies performed to study implantation in the uterus. Therefore, we agree with this reviewer that intrauterine embryonic development is affected by superovulation even in WT mice. Accordingly, our studies using superovulation were intended only to be used for the study of ovarian function, and not intrauterine embryonic development. Further analysis of CLs in the *Acvr2a* cKO mice were not performed because our focus was to determine ACVR2A function in the uterus during embryo implantation. Future studies to investigate the molecular contribution of ACVR2A to CL function in the ovary (if there are any) are beyond the scope of this manuscript.

To clarify this point, we have updated the Results and Discussion sections to address this reviewer's comment.

Results Section Lines 290-293:

*"To study the morphology of the ovary, mice were induced to superovulate with PMSG + hCG for 6 hours (to assess pre-ovulatory follicles). Morphological analysis of the ovaries revealed both normally developing follicles and follicles with defective cumulus cells (Supplementary Fig. 5D-E). Corpora lutea were analyzed 18 hours after PMSG + hCG administration, and analyses showed the presence of corpora lutea in the ovaries of both control and *Acvr2a* cKO mice (Supplementary Fig. 5F-G)."*

Discussion Section Lines 458-461:

*"Analyses of ovarian function, independent of uterine function, were performed in these mouse models by administering PMSG + hCG. Ovulation rate and ovarian architecture was normal in *Smad1/5* cKO and *Acvr2b* cKO mice and subtle defects were observed in the ovaries of *Acvr2a* cKO mice."*

Did the ICI-treated mice show normal implantation in WT mice on day 8?

Implantation sites from mice in the ICI rescue experiment (new Supplementary Figure 3) were assessed at 5.5dpc. This timepoint was chosen to determine whether the anti-estrogenic effect of ICI administration was sufficient to rescue implantation, which in mice occurs at 4.5dpc. Therefore, while we did not assess embryo development beyond 5.5dpc, previous studies (i.e., PMID 20219888) have identified that pregnancy can progress to term in WT mice treated with ICI. Thus, we expect that implantation sites assessed in ICI-treated WT mice at day 8 will be developmentally normal and comparable to the implantation sites of vehicle-treated mice. To reflect this, we have clarified why 5.5dpc was used as the timepoint to analyze implantation in the Results (Lines 246-247) and Methods sections. Lines 678-679 now state,

"Because implantation in the mouse occurs at 4.5dpc, implantation was assessed in the afternoon of 5.5dpc using injection of Sky Blue Dye."

Reviewer #2 (Remarks to the Author):

The authors have extensively revised the manuscript and have fully addressed all of my concerns. The flow is more logical and human endometrium/implantation is appropriately discussed. I have one suggestion and two very minor amendment requests.

We thank the reviewer for this favorable review of our edited manuscript and are delighted that they find the changes have improved the quality and flow of the paper.

1. I would suggest placing less emphasis on the role of E2 in the abstract, given that E2 antagonism did not affect implantation in controls and did not rescue implantation in genetically altered models.

We appreciate this suggestion and have edited the abstract to remove emphasis from the activity of E2.

2. Line 93, please change 'both endometrial cell types' to 'Two endometrial cell types'. The endometrium is multicellular and its composition dynamic, changing across the cycle.

We thank the reviewer for pointing this out. We have updated this line in the introduction as suggested. The line now states,

“In the endometrium, various cell types are critical in establishing a pregnancy, endometrial epithelial cells participate in maternal-embryonic communication during implantation, and stromal cells transform into a secretory cell type (i.e., decidualize), with the important role of nurturing the growth and development of the early embryo”

3. Line 112-3 'In the mouse uterus'

Thank you. This sentence has been edited and now reads,

“In the mouse uterus, in vivo studies have shown that conditional deletion of BMP2 or ALK2 results in female infertility due to defects in the post-implantation process of stromal cell decidualization.”